# Divergent *Plasmodium* kinases drive MTOC, kinetochore and axoneme organisation in male gametogenesis

Ryuji Yanase[1,2] , Mohammad Zeeshan[1], David JP Ferguson[3,4] , Robert Markus[1] , Declan Brady[1] , Andrew R Bottrill[5] , Anthony A Holder[6] , David S Guttery[1,2] , Rita Tewari[1]

Sexual development and male gamete formation of the malaria parasite in the mosquito midgut are initiated by rapid endomitosis in the activated male gametocyte. This process is highly regulated by protein phosphorylation, specifically by three divergent male-specific protein kinases (PKs): CDPK4, SRPK1, and MAP2. Here, we localise each PK during male gamete formation using live-cell imaging, identify their putative interacting partners by immunoprecipitation, and determine the morphological consequences of their absence using ultrastructure expansion and transmission electron microscopy. Each PK has a distinct location in either the nuclear or the cytoplasmic compartment. Protein interaction studies revealed that CDPK4 and MAP2 interact with key drivers of rapid DNA replication, whereas SRPK1 is involved in RNA translation. The absence of each PK results in severe defects in either microtubule-organising centre organisation, kinetochore segregation, or axoneme formation. This study reveals the crucial role of these PKs during endomitosis in formation of the flagellated male gamete and uncovers some of their interacting partners that may drive this process.

## Introduction

Mitotic cell division in the malaria parasite, *Plasmodium*, is characterised by two forms of atypical, closed mitosis. The first is schizogony in hepatocytes and red blood cells of the mammalian host and sporogony within oocysts in the mosquito gut. The second is in development of activated male gametocytes within the mosquito midgut during sexual development to form eight haploid, flagellated, motile male gametes. Closed mitosis during male gametogenesis involves three rapid rounds of DNA replication over 8–10 min in an endomitotic division to produce an octoploid nucleus. Only once all rounds of nuclear division are complete does exflagellation occur to produce male gametes (Sinden, 2015; Guttery et al, 2022).

Reversible protein phosphorylation, catalysed by protein kinases (PKs) and protein phosphatases (PPs), is a ubiquitous, highly conserved mechanism of protein activation/deactivation that is known to regulate mitosis in *Eukaryota* (Nilsson, 2019; Strumillo et al, 2019; Fulcher & Sapkota, 2020). Protein components of key structures in mitosis, such as microtubule-organising centres (MTOCs), spindles, and kinetochores, are subject to spatiotemporal regulation mediated by the coordinated activity of various PKs and PPs to enable accurate cell cycle control and chromosome segregation (Ito & Bettencourt-Dias, 2018; Ong et al, 2020; Vagnarelli, 2021; Kucharski et al, 2022). In the rodent malaria parasite *Plasmodium berghei* (Pb), several studies have highlighted three male-specific PKs, which are defined as essential during male gametogenesis but not at the asexual blood stage. These are calcium-dependent protein kinase-4 (CDPK4), a serine/arginine-rich protein kinase (SRPK1), and mitogen-activated protein kinase-2 (MAP2; Billker et al, 2004; Tewari et al, 2005; Tewari et al, 2010; Guttery et al, 2024). There are significant differences in sequence, domain architecture, predicted substrate specificity, and function between these PKs in *Plasmodium* and their orthologues in other eukaryotes, so they are considered to be divergent in *Plasmodium* (Tewari et al, 2005; Billker et al, 2009; Dixit et al, 2010). The deletion of each gene ablates exflagellation (male gamete release) and completely blocks sexual reproduction and parasite transmission through the mosquito. In contrast, the deletion of each PK has little to no effect on the parasites' development during the asexual blood stage (Billker et al, 2004; Tewari et al, 2005, 2010). Each PK has been postulated to act at a distinct point: CDPK4 is considered to be a master regulator, initiating assembly of the pre-replicative complex and formation of the first mitotic spindle within the first 18 s after gametocyte activation (Billker et al, 2004; Fang et al, 2017; Invergo et al, 2017); SRPK1 is required for subsequent DNA replication (Invergo et al, 2017), and MAP2 is required for cytokinesis and axoneme motility (Tewari et al, 2005; Guttery et al, 2012).

[1]School of Life Sciences, University of Nottingham, Nottingham, UK   [2]Department of Genetics, Genomics and Cancer Sciences, University of Leicester, Leicester, UK   [3]Nuffield Department of Clinical Laboratory Sciences and John Radcliffe Hospital, University of Oxford, Oxford, UK   [4]Department of Biological and Medical Sciences, Faculty of Health and Life Sciences, Oxford Brookes University, Oxford, UK   [5]School of Life Sciences, Gibbet Hill Campus, University of Warwick, Coventry, UK   [6]Malaria Parasitology Laboratory, Francis Crick Institute, London, UK

Correspondence: dsg6@leicester.ac.uk; rita.tewari@nottingham.ac.uk

The MTOC is crucial for the completion of DNA replication (Ito & Bettencourt-Dias, 2018; Guttery et al, 2022). The MTOC is known in yeast as the spindle pole body, in humans as the centrosome, and in ciliated cells as the basal body (Seybold & Shiebel, 2013; Vaughan & Gull, 2016; Vertii et al, 2016; Ito & Bettencourt-Dias, 2018). Although there are similar centrosome-like structures in other apicomplexans, the formation of the MTOC during *Plasmodium* male gametogenesis is unique, consisting of a paired bipartite structure spanning the nuclear membrane (Zeeshan et al, 2022a; Rashpa & Brochet, 2022). This structure serves in the nucleus as an inner acentriolar MTOC (also known as a nuclear pole) for spindle microtubules and in the cytoplasm as an outer centriolar MTOC where the basal body and axoneme are assembled (we refer to the inner and outer MTOCs as "spindle MTOC" and "basal body MTOC," respectively). The bipartite MTOC is assembled, duplicated, and separated at each round of mitosis, resulting in the formation of eight flagellated haploid male gametes (Guttery et al, 2022). Several PKs participate in MTOC function (Vertii et al, 2016); for example, in humans and *Drosophila*, Aurora kinase A and polo-like kinase 1 (Plk1) are recruited to the spindle poles and play a crucial role in spindle assembly and mitotic progression (Barr & Gergely, 2007). Also, never in mitosis (NIMA)–like kinases (NEKs) and cyclin-dependent kinases (CDKs) are known regulators of MTOC biology in diverse organisms, including Aspergillus, plants, yeast, and human cells (Joubes et al, 2000; Elserafy et al, 2014; Fry et al, 2017; Panchal & Evan Prince, 2023). In apicomplexan parasite *Toxoplasma gondii*, Aurora, NEK, and CDK-related kinase (CRK) are located at mitotic structures and play crucial roles in cell cycle regulation (Chen & Gubbels, 2013; Suvorova et al, 2015; Gaji et al, 2021). *Plasmodium* NEK1 kinase is an important component of MTOC organisation and an essential regulator of chromosome segregation during male gamete formation (Zeeshan et al, 2024). *Plasmodium* CRK5, another divergent male-specific PK, is a critical protein kinase that regulates male gametogenesis (Balestra et al, 2020; Kumar et al, 2022). Aurora kinase B is a known essential regulator of chromosome segregation and cytokinesis in organisms ranging from yeast to humans (Honda et al, 2003; Carmena et al, 2009, 2015; Hadders & Lens, 2022), and the highly divergent paralogue, *Plasmodium* ARK2, is essential for spindle dynamics and chromosome segregation during male gametogenesis (Zeeshan et al, 2023). However, the involvement of other *Plasmodium* divergent male-specific kinases in chromosome segregation is not well understood.

Both CDPK4 and SRPK1 may play early roles in homeostasis of gametocyte bipartite MTOC and axoneme formation (Zeeshan et al, 2022a; Rashpa & Brochet, 2022). In *cdpk4* deletion mutants, axoneme formation is severely impaired, with only short, incomplete axonemes being formed, whereas *srpk1* deletion mutant parasites have incorrectly positioned basal bodies (Rashpa & Brochet, 2022). In contrast, a *map2* mutant line forms normal axonemes but has a severe defect in chromosome condensation (Guttery et al, 2012). It remains unclear which axoneme-associated proteins are phosphorylated by these PKs, and what their specific roles are during axoneme formation.

We aimed to localise each PK in relation to MTOC assembly, axoneme formation, and chromosome segregation, as well as identify some of their interacting partner proteins—particularly for MAP2 and SRPK1 for which little is known. Using a combination of live-cell imaging, protein identification by proteomics, and ultrastructure analysis by expansion and electron microscopy, we show that PbCDPK4 and PbSRPK1 are diffusely distributed throughout the parasite cell, with a slightly stronger punctate cytoplasmic localisation during schizogony and male gametogenesis, whereas PbMAP2 is preferentially present in the nucleus and during male gametogenesis. These three PKs interact with each other and with other key regulators of MTOC biology and male gametogenesis. The ultrastructure analysis revealed that their absence affects MTOC formation, duplication, and separation, kinetochore dynamics, and axoneme development.

## Results and Discussion

### CDPK4, SRPK1, and MAP2 have different spatiotemporal locations during schizogony and male gametogenesis

Despite several studies highlighting the essential roles of CDPK4, SRPK1, and MAP2 in male gametogenesis in *P. berghei* (Billker et al, 2004; Tewari et al, 2010), little is known about their location during the endomitotic stages of cell division (i.e., schizogony, sporogony, and male gametogenesis). To examine the spatiotemporal expression of CDPK4, SRPK1, and MAP2, we generated transgenic *P. berghei* parasite lines expressing the genes modified to code for a C-terminal GFP-tag. An in-frame *gfp* coding sequence was inserted at the 3′ end of the endogenous gene locus (*cdpk4*—PBANKA_0615200; *srpk1*—PBANKA_0401100; *map2*—PBANKA_0933700), using single-crossover homologous recombination (Fig S1A, Table S1 for primers), and successful insertion was confirmed by diagnostic PCR (Fig S1B, Table S1). Western blot analysis of schizont (for CDPK4 and SRPK1) and gametocyte (for MAP2 as MAP2 seems to be barely expressed in asexual blood stages) protein extracts, using an anti-GFP antibody, revealed major bands at 87.6 kD for CDPK4-GFP, 180.7 kD for SRPK1-GFP, 88.1 kD for MAP2-GFP, and 27 kD for unfused GFP (WT-GFP), respectively; the expected sizes for each. Each GFP-tagged PK was also observed by fluorescence in activated gametocytes (Fig S1C).

Live-cell fluorescence imaging of *P. berghei* asexual blood stages revealed a diffuse cytoplasmic distribution of CDPK4-GFP in both schizonts and merozoites. In contrast, SRPK1-GFP showed weak diffuse cytoplasmic distribution with a strong focus at the location of the nuclear pole and towards the apex of the forming or mature merozoites (Fig 1A). In contrast, no MAP2-GFP fluorescence was observed in asexual development. During male gametogenesis, CDPK4-GFP had a diffuse location in both cytoplasm and nucleus, along with a distinct focus in the cytoplasm six min post-activation (Fig 1B, arrowheads). At the early stage of male gametogenesis (30 s to 6 min post-activation), up to two foci of CDPK4-GFP were observed, whereas at the late stage (>6 min post-activation), more than three foci were sometimes observed (Fig S1D, arrowheads). SRPK1-GFP was located in the cytoplasm and was largely excluded from the nucleus, confirming previous observations in *Plasmodium falciparum* (Pf; Dixit et al, 2010; Kumar et al, 2021). MAP2-GFP is preferentially located in the nucleus in activated male gametocytes (Fig 1B).

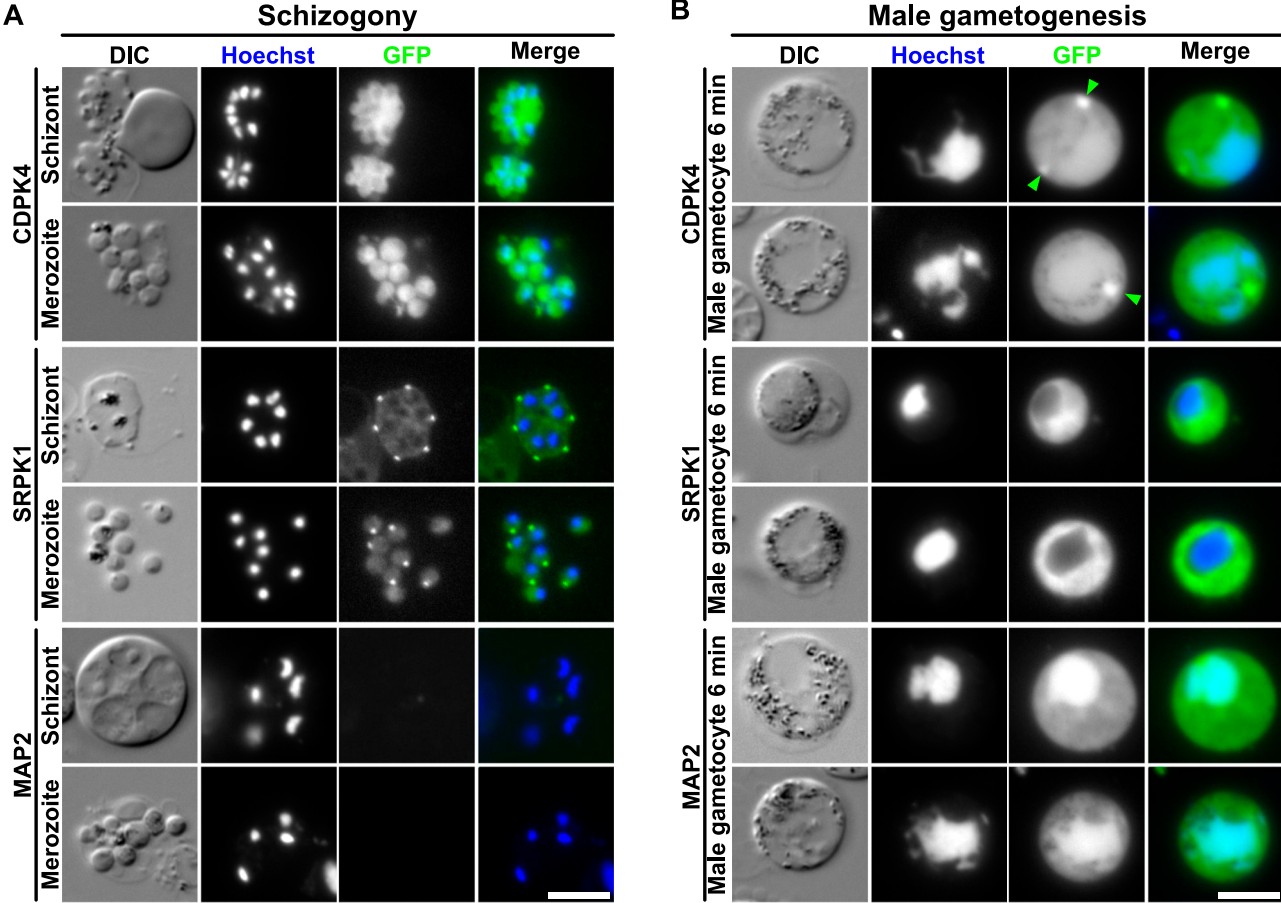

**Figure 1.  Fluorescence localisation in PK-tagged lines during asexual and sexual stages.**
**(A)** Localisation of each GFP-tagged PK in schizonts or merozoites. DIC: differential interference contrast. Merge shows Hoechst (DNA, blue) and GFP (green) signals. Scale bar = 5 μm. **(B)** Localisation of each GFP-tagged PK in activated gametocytes 6 min post-activation. Merge shows Hoechst (DNA, blue) and GFP (green) signals. Green arrowheads indicate CDPK4-GFP concentrated focus in the cytoplasm. Scale bar = 5 μm.

Because these PKs are essential for axoneme development and have specific locations in the parasite cell (in particular, the distinct CDPK4-GFP focus in male gametocytes), we examined their co-localisation with markers of MTOC biology, including chromosome segregation (kinetochore protein, NDC80; Zeeshan et al, 2020) and basal body formation (cytoplasmic axonemal protein, Kinesin-8B; Zeeshan et al, 2019), which were C-terminally tagged with mCherry (mCh). No co-localisation was observed between CDPK4-GFP, SRPK1-GFP, or MAP2-GFP and NDC80-mCh during male gametogenesis (Fig 2A), or between SRPK1-GFP and NDC80-mCh during schizogony (Fig S2A), suggesting that these PKs do not directly interact with kinetochores during chromosome condensation. However, during male gametogenesis, SRPK1-GFP appeared to weakly co-localise with Kinesin-8B-mCh in some areas (Figs 2B and S3). CDPK4-GFP did not co-localise with Kinesin-8B-mCh at any point (Fig 2B). This was surprising, because altered Kinesin-8B phosphopeptide abundance has been observed in *cdpk*4 mutants (Fang et al, 2017; Kumar et al, 2021). A MAP2-GFP/Kinesin-8B-mCh cross was not generated because these proteins are primarily located in different subcellular compartments.

## CDPK4, SRPK1, and MAP2 interact with each other and key regulators of male gametogenesis

Previous studies have highlighted differential protein phosphorylation, detected by altered abundance of phosphopeptides in protease digests, of cellular proteins associated with distinct cell cycle events in Pb*cdpk4* and Pb*srpk1* deletion mutants (Fang et al, 2017; Invergo et al, 2017). In addition, SRPK1 was suggested to be phosphorylated in a CDPK4-dependent manner (Invergo et al, 2017). Fang et al (2017) identified CDPK4-interacting proteins and CDPK4-dependent phosphorylation, using immunoprecipitation and mass spectrometry–based phosphoproteomics. However, the protein substrates of SRPK1 and MAP2 during male gametogenesis are largely unknown. We analysed the interactome of each PK in *P. berghei* gametocyte lysates to identify interacting partners of each PK, which are potential substrates and may regulate male gametogenesis.

In triplicate experiments, we immunoprecipitated with anti-GFP antibody CDPK4-GFP, SRPK1-GFP, MAP2-GFP, and WT-GFP (as a control) from lysates of cells, 6 min post-activation of gametocytes. Immunoprecipitates were digested, and the resultant peptides

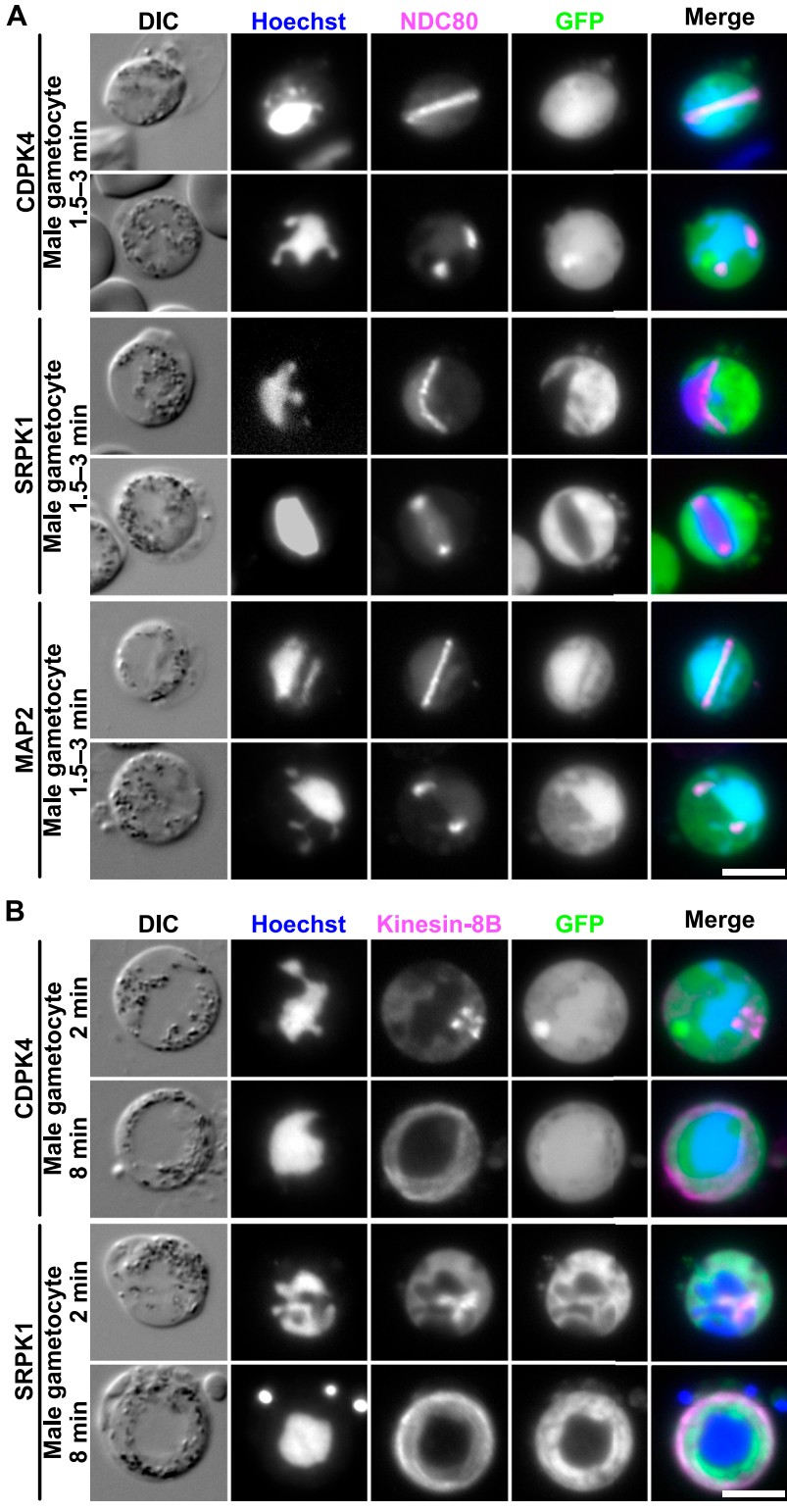

**Figure 2. Co-localisation of PKs with kinetochore and basal body/axoneme markers.**
**(A)** Co-localisation of each GFP-tagged PK with mCherry (mCh)-tagged NDC80 in gametocytes, 1.5–3 min post-activation. DIC: differential interference contrast. Merge shows Hoechst (DNA, blue), NDC80-mCh (magenta), and GFP (green). Scale bar = 5 μm.
**(B)** Co-localisation of each GFP-tagged PK with mCh-tagged Kinesin-8B in gametocytes 2–8 min post-activation. Merge shows Hoechst (DNA, blue), Kinesin-8B-mCh (magenta), and GFP (green). Scale bar = 5 μm.

were analysed by mass spectrometry to identify component proteins. Potential interacting partners for each PK were identified as those with unique peptide sequences present in all three biological replicates, but completely absent from WT-GFP control samples.

A total of 108, 201, and 162 proteins were immunoprecipitated together with CDPK4-GFP, SRPK1-GFP, and MAP2-GFP, respectively (Fig 3A, Tables 1, S2, S3, and S4). Gene Ontology (GO) analysis of CDPK4- and MAP2-GFP partners highlighted several components of

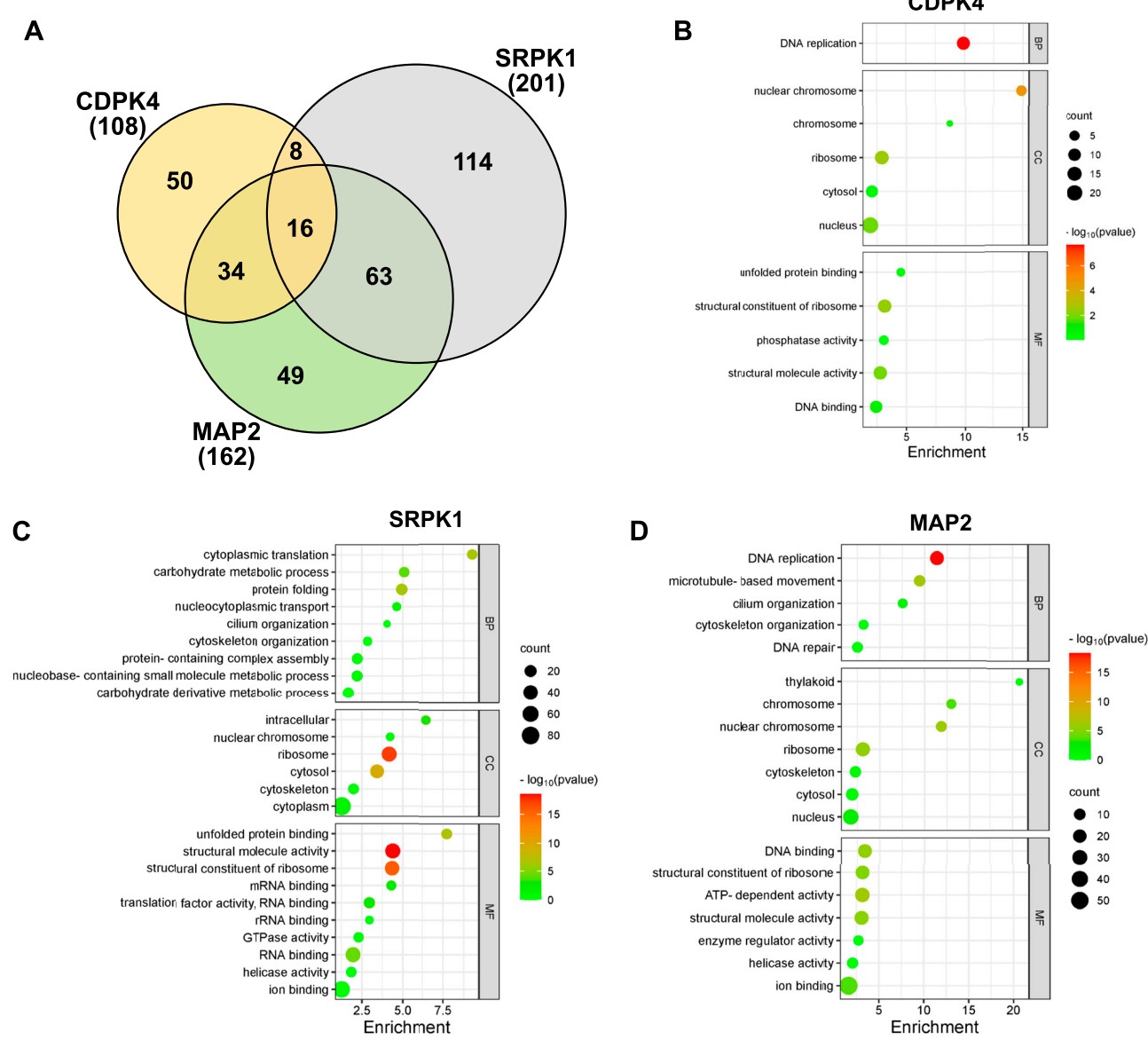

**Figure 3. Interactome of CDPK4, SRPK1, and MAP2 during male gametogenesis.**
**(A)** Venn diagram showing common interacting partners in gametocytes activated for 6 min, with some additional proteins specific to PK. Each number represents the number of identified proteins. **(B, C, D)** GO enrichment analysis of CDPK4-GFP (B), SRPK1-GFP (C), and MAP2-GFP (D). BP, biological process; CC, cellular component; MF, molecular function. −log10 *P*-values obtained from the Bonferroni-adjusted *P*-values. See also Tables 1, S2, S3, S4, S5, S6, and S7.

the replisome and DNA replication (Fig 3B and D, Tables S5, S6, and S7). Of note, in the CDPK4-GFP line all five known *Plasmodium* origin recognition complex (ORC) subunits and CDC6 (components of the pre-replicative complex; Chou et al, 2021) were present (Tables 1 and S2), along with the catalytic subunits of DNA polymerase alpha and epsilon. However, no minichromosome maintenance (MCM) proteins were detected, unlike in the MAP2-GFP samples, in which MCM3–7 were detected (Tables 1 and S4). These data are consistent with a role of CDPK4 in activation of the pre-replicative complex (Invergo et al, 2017), and this role may explain the complete lack of DNA replication in *cdpk4* deletion mutants (Tewari et al, 2010). In higher eukaryotes, phosphorylation of ORCs is facilitated by CDKs (Lee et al, 2012), whereas Cdc7-Dbf4 PK (DDK) promotes the

assembly of a stable Cdc45-MCM complex (Sheu & Stillman, 2006). No homologues of either Cdc7, Dbf4, or Cdc45 have been identified in the *Plasmodium* genome, and therefore, we suggest that MAP2 performs the function of DDK. MCM2, MCM4, and MCM7 were also co-precipitated with SRPK1-GFP (Tables 1 and S3). Live-cell imaging of male gametocytes suggested that SRPK1-GFP is largely excluded from the nucleus (Fig 1B), but by immunoprecipitation, we detected interactions between SRPK1-GFP and proteins known to be generally found in the nucleus, such as MCM proteins. MCM proteins shuttle between the nucleus and cytoplasm, and it has been suggested that they interact with protein kinases and protein phosphatases in the cytoplasm (Abe et al, 2012). Thus, an interaction between SRPK1-GFP and MCM proteins may occur in the

**Table 1.  Binding partners and potential substrates of male-specific protein kinases during male gametogenesis.**

| | P. berghei gene ID | Product description | Gene names | Average no. Unique peptides | | |
|---|---|---|---|---|---|---|
| | | | | CDPK4-GFP | SRPK1-GFP | MAP2-GFP |
| **Reversible protein phosphorylation** | PBANKA_0615200 | Calcium-dependent kinase 4 | CDPK4 | 42 | 6 | 10 |
| | PBANKA_0401100 | Serine/threonine protein kinase, putative | SRPK1 | 5 | 66 | 0 |
| | PBANKA_0933700 | Mitogen-activated protein kinase 2 | MAP2 | 5 | 0 | 28 |
| | PBANKA_1227400 | Serine/threonine protein phosphatase 2B catalytic subunit A | CNA | 3 | 0 | 0 |
| | PBANKA_1131900 | Serine/threonine protein phosphatase 5 | PP5 | 4 | 0 | 0 |
| | PBANKA_0607400 | Protein phosphatase PPM11, putative | PPM11 | 20 | 0 | 13 |
| | PBANKA_0602000 | Origin recognition complex subunit 1, putative | ORC1 | 24 | 0 | 0 |
| | PBANKA_0803000 | Origin recognition complex subunit 2, putative | ORC2 | 23 | 0 | 8 |
| | PBANKA_0513900 | ORC3 domain–containing protein, putative | ORC3 | 23 | 0 | 5 |
| | PBANKA_1348800 | Origin recognition complex subunit 4, putative | ORC4 | 16 | 0 | 0 |
| | PBANKA_0312500 | Origin recognition complex subunit 5, putative | ORC5 | 19 | 0 | 6 |
| **DNA replication** | PBANKA_1102900 | Cell division control protein 6, putative | CDC6 | 12 | 0 | 9 |
| | PBANKA_1024900 | DNA replication licensing factor MCM2, putative | MCM2 | 0 | 5 | 0 |
| | PBANKA_1241800 | DNA replication licensing factor MCM3, putative | MCM3 | 0 | 0 | 21 |
| | PBANKA_1415600 | DNA replication licensing factor MCM4, putative | MCM4 | 0 | 3 | 27 |
| | PBANKA_0610200 | DNA replication licensing factor MCM5, putative | MCM5 | 0 | 0 | 18 |
| | PBANKA_1131600 | DNA replication licensing factor MCM6, putative | MCM6 | 0 | 0 | 17 |
| | PBANKA_0803100 | DNA replication licensing factor MCM7, putative | MCM7 | 0 | 3 | 22 |
| | PBANKA_1428700 | Eukaryotic translation initiation factor 3 subunit A, putative | eIF3A | 0 | 14 | 0 |
| | PBANKA_1232500 | Eukaryotic translation initiation factor 3 subunit B, putative | eIF3B | 0 | 9 | 0 |
| | PBANKA_0604800 | Eukaryotic translation initiation factor 3 subunit C, putative | eIF3C | 0 | 9 | 0 |
| | PBANKA_1206100 | Eukaryotic translation initiation factor 3 subunit D, putative | eIF3D | 0 | 13 | 0 |
| | PBANKA_1242800 | Eukaryotic translation initiation factor 3 subunit E, putative | eIF3E | 4 | 3 | 3 |
| **Translation/mRNA splicing** | PBANKA_0715200 | Eukaryotic translation initiation factor 3 subunit G, putative | eIF3G | 0 | 4 | 0 |
| | PBANKA_0614500 | Eukaryotic translation initiation factor 3 subunit I, putative | eIF3I | 0 | 6 | 0 |
| | PBANKA_0720300 | Eukaryotic translation initiation factor 3 subunit M, putative | eIF3M | 0 | 5 | 0 |
| | PBANKA_0711900 | Heat shock protein 70 | HSP70 | 0 | 37 | 0 |
| | PBANKA_0805700 | Heat shock protein 90, putative | HSP90 | 0 | 32 | 0 |
| | PBANKA_0610900 | HSP40, subfamily A, putative | HSP40 | 0 | 27 | 0 |

**Table 1. Continued**

| | P. berghei gene ID | Product description | Gene names | Average no. Unique peptides | | | |
|---|---|---|---|---|---|---|---|
| | | | | CDPK4-GFP | SRPK1-GFP | MAP2-GFP | |
| | PBANKA_1416900 | Structural maintenance of chromosomes protein 2 | SMC2 | 7 | 0 | 11 | |
| | PBANKA_1108700 | Structural maintenance of chromosomes protein 4 | SMC4 | 12 | 0 | 18 | |
| | PBANKA_0917500 | Structural maintenance of chromosomes protein 1, putative | SMC1 | 0 | 7 | 8 | |
| | PBANKA_0716000 | Structural maintenance of chromosomes protein 3, putative | SMC3 | 0 | 6 | 5 | |
| Chromosome organisation/MTOC | PBANKA_0917400 | Armadillo repeat protein PF16 | PF16 | 0 | 6 | 8 | |
| | PBANKA_0202700 | Kinesin-8B, putative | Kinesin-8B | 0 | 12 | 14 | |
| | PBANKA_1458300 | Kinesin-13, putative | Kinesin-13 | 6 | 0 | 6 | |
| | PBANKA_1458800 | Kinesin-15, putative | Kinesin-15 | 0 | 0 | 8 | |
| | PBANKA_1310400 | Centrin-2, putative | Centrin-2 | 0 | 6 | 0 | |

List of proteins interacting with CDPK4-GFP, SRPK1-GFP, and MAP2-GFP during male gametogenesis 6 min post-activation. Average unique peptides represent an average across three biological replicates.

cytoplasm. Phosphorylation of MCM2 is critical for its loading onto chromatin (Chuang et al, 2009), so it is possible that CDPK4 is critical for the association of the ORC with chromatin and the replication origin, whereas SRPK1 and MAP2 facilitate activation of the MCM complex. CDPK4 is also thought to function not only as a kinase but also as a scaffold protein, forming complexes with other proteins such as ORC subunits or CDC6 (Fang et al, 2017), and promoting the interaction between a phosphorylation product and its binding protein (Popova et al, 2018). An altered phosphopeptide profile was only observed in the *srpk1* deletion mutant for MCM5 and in the *cdpk4* mutant for ORC1 (Invergo et al, 2017), so our hypothesis differs somewhat from that of Fang et al (2017), who identified CDPK4 as part of the MCM complex in non-activated gametocytes. However, in the current study gametocytes were analysed 6 min post-activation, a time point at which MCM function may be regulated by SRPK1 or MAP2, rather than CDPK4.

MAP2-GFP complexes were also enriched with several MTOC-associated proteins involved in axoneme development including Kinesin-8B, Kinesin-13, and Kinesin-15 and PF16 (Fig 3D, Tables 1 and S4; Straschil et al, 2010; Zeeshan et al, 2019, 2022b). Our results are in partial agreement with those of Fang et al (2017); Kumar et al (2021), who found altered phosphopeptide abundance for all three kinesins in both Pb*cdpk4* and Pb*srpk1* deletion mutants (Fang et al, 2017), and Pf*cdpk4* mutants (Kumar et al, 2021), respectively. In addition, SMC2/4 was co-precipitated with both CDPK4 and MAP2, whereas SMC1/3 was co-precipitated with SRPK1 and MAP2. SMC1 and SMC3 are part of the cohesin complex that facilitates sister-chromatid cohesion during mitosis, whereas SMC2/4 are part of the condensin complex that regulates chromosome condensation and segregation during cell division (Uhlmann, 2016). Knockdown of *smc2/4* significantly affects male exflagellation and parasite transmission (Pandey et al, 2020). In yeast, SMC4 activity and dynamic binding of the condensation machinery are governed by Cdk1 (Robellet et al, 2015); thus, it is possible that a major regulator of condensin activity may be CDPK4 because it interacts with SMC4 (Fang et al, 2017). Our study also suggests that SMC2/4 play a role late in male gametogenesis through interaction with MAP2-GFP, possibly a role in atypical chromosome condensation through association with the MTOC (Guttery et al, 2012).

SRPK1-GFP precipitates were strongly enriched for the translation pathway and 80S ribosome proteins, including 18 40S and 20 60S ribosomal proteins, and several subunits of the eukaryotic translation initiation factor 3 (eIF3) complex (Fig 3C, Tables 1 and S6). mRNA translation, translation elongation, and termination are highly coordinated processes facilitated by the 80S ribosome and an extensive array of eIFs, including eIF3 (Kramer et al, 2019). Co-precipitation of these elements was not unexpected because of SRPK1's known role in mRNA splicing and translation (Dixit et al, 2010) and suggests that this process, including phosphorylation of the eIF3 complex (Farley et al, 2011), is highly conserved in Apicomplexa. SRPK1 also co-precipitated the co-chaperone HSP40, a protein that mediates dynamic interactions of SRPK1 with the major molecular chaperones HSP70 and HSP90 in mammalian cells (Zhong et al, 2009), and proteins that were detected here (Table 1). However, unlike the findings of other studies (Dixit et al, 2010), we observed no interaction of SRPK1 with serine/arginine-rich (SR) proteins.

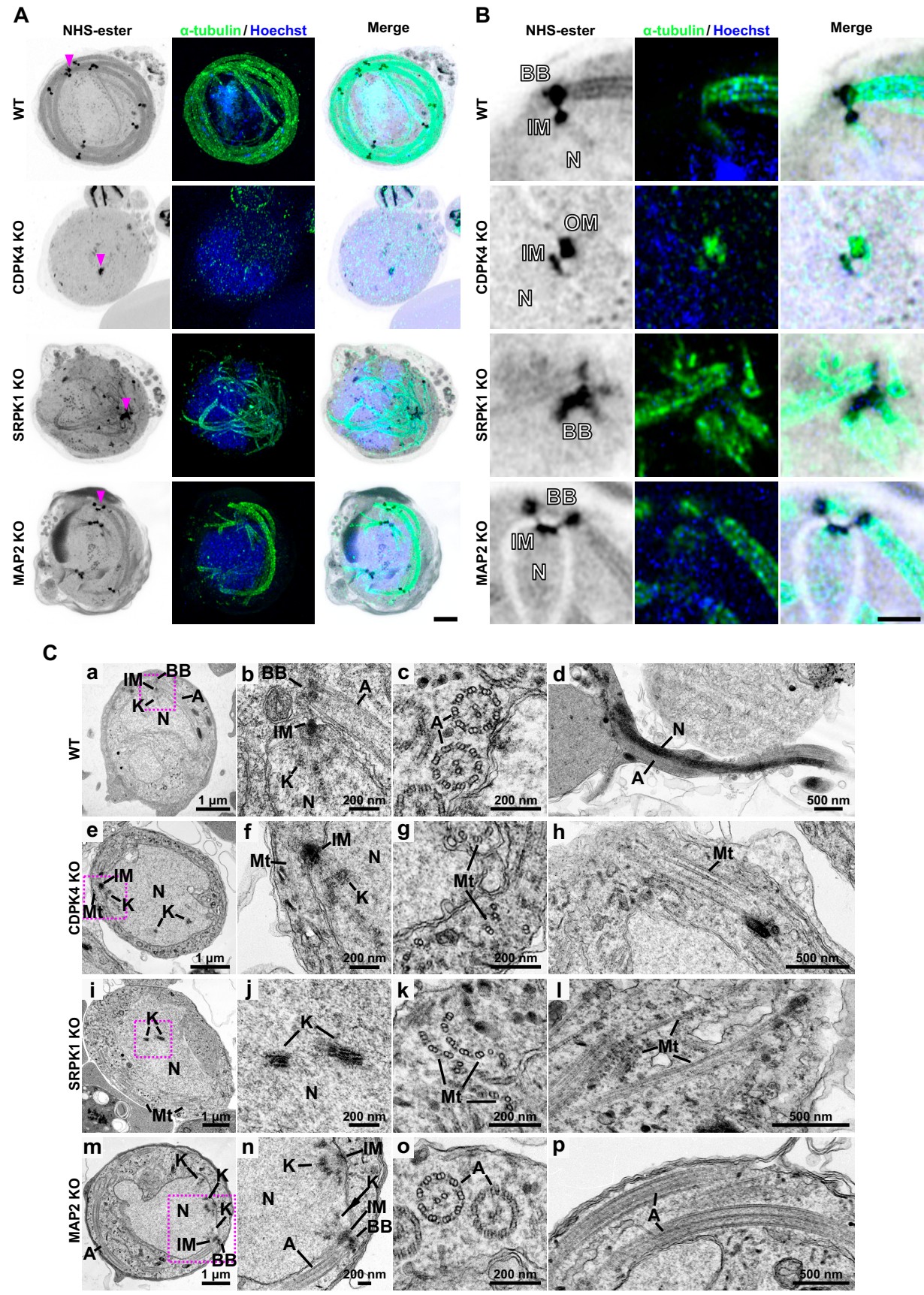

Finally, our data are consistent with a role of reversible protein phosphorylation as a key driver of male gametogenesis. CDPK4 and SRPK1 co-precipitated each other (Table 1), consistent with a potential feedback mechanism between these PKs, as suggested previously (Invergo et al, 2017). We also observed co-precipitation of CDPK4 and MAP2 with PPM11, a novel PP shown previously to play a role in male gametogenesis (Balestra et al, 2020). As we also found, altered phosphopeptide abundance was observed for PPM11 in *cdpk4* but not *srpk1* mutants (Fang et al, 2017), which suggests that PPM11 has a role in mitotic entry and exit through dephosphorylation of CDPK4 and MAP2, respectively. CDPK4 function may also be regulated by calcineurin and PP5 (Table 1), both of which have been implicated in male gametogenesis (Philip & Waters, 2015; Zhu et al, 2019).

## Ultrastructure analysis reveals defects in MTOC formation, kinetochore dynamics, and axoneme formation in PK mutants

To determine the ultrastructural defects resulting from loss of these PKs, we used male gametocytes of *cdpk4*, *srpk1*, and *map2* deletion mutant lines (Tewari et al, 2010) activated for 8 min and examined using ultrastructure expansion microscopy (U-ExM; Gambarotto et al, 2019), and activated for 8 or 15 min and examined using transmission electron microscopy (Fig 4).

In the *cdpk4* mutant, U-ExM revealed a single bipartite MTOC in the nucleus, together with a short microtubular structure extending from the MTOC (Figs 4A and B and S4A and B). A similar single bipartite MTOC was observed by U-ExM of CDPK4-KO male gametocytes in a previous study, and centrin was shown to localise to the outer basal body MTOC (Rashpa & Brochet, 2022). Normal development and location of the bipartite MTOC, basal bodies, kinetochores, axonemes, and microtubules revealed in WT parasites by electron microscopy (Fig 4Ca–d) were disrupted in the *cdpk4* mutant with some kinetochores appearing to remain within the nucleoplasm (Fig 4Ce), and several attached to the spindle radiating from the MTOC (Fig 4Cf). Dispersed doublet microtubules were present in the cytoplasm, which are likely precursors of axonemes that had failed to form (Figs 4Cf–h and S5C, D, G, and H). These results are like those of Rashpa & Brochet (2022), who had suggested that *cdpk4* mutants form partial microtubules from the MTOC. Although CDPK4 interacts with multiple proteins, this phenotype is reminiscent of that of a Kinesin-13–deficient line (Zeeshan et al, 2022b), consistent with our finding that CDPK4 interacts with Kinesin-13 (Table 1). However, our immunoprecipitation data show that CDPK4 interacts with multiple proteins, and its phosphorylation cascade is considered to be complex (Invergo et al, 2017). Therefore, not only CDPK4 and Kinesin-13, but also multiple other proteins may interact to enable spindle assembly, DNA segregation,

and axoneme formation during male gametogenesis as suggested in previous studies (Fang et al, 2017; Invergo et al, 2017; Kumar et al, 2022).

The *srpk1* mutant lacked MTOCs associated with the nucleus (Figs 4A and S4A and B), and kinetochores were located near the centre of the nucleoplasm (Figs 4Ci and j and S5I, J, M, and N). Whereas in WT cells, all kinetochores (29 of 29 kinetochores from 12 cells) were associated with spindles, in SRPK1-KO cells, no kinetochores (0 of 20 kinetochores from 10 cells) were associated with spindles and all kinetochores were separated from the nucleus periphery (Figs 4Ci and j and S5I, J, M, and N). In SRPK1-KO cells, abnormal basal bodies formed clusters away from the nucleus (Figs 4A and B and S4A), together with partially formed, abnormal axonemes lacking the usual 9+2 WT tubulin subunit configuration (Figs 4Cc, d, k, and l and S5K, L, O, and P). Similar observations were made by Rashpa & Brochet (2022), who found that CDPK4 and SRPK1 are key for axoneme assembly, microtubule nucleation, and development of the MTOC. The loss of the connection between the spindle and the basal body/outer MTOC is a phenotype also observed in *Toxoplasma* tachyzoites after the knockout or knockdown of the kinetochore and its associated proteins (Farrell & Gubbels, 2014; Brusini et al, 2022; Li et al, 2024). However, the unique MTOC formation and function in *Plasmodium* gametocytes are likely achieved through complex interactions between various proteins, including kinetochore components and PKs such as CDPK4 and SRPK1.

We showed previously that *map2* deletion results in defective chromosome condensation with arrested kinetochore development during male gametogenesis (Guttery et al, 2012). In a *map2* mutant line, the two bipartite MTOCs remained adjacent without separation (62% [21/34] of MTOCs from six cells; Figs 4A–Cn and S4A and B), whereas in WT cells, they were dispersed throughout the cytoplasm (100% [28/28] of MTOCs from five cells; Fig 4A–Ca and b). Axoneme structure in a MAP2-KO line appeared to be similar to that in WT cells (Fig 4A and B), with the classical 9+2 subunit arrangement (Figs 4Cm–p and S5S, T, W, and X). However, despite the formation of apparent intact axonemes, exflagellation was not observed. We suggested previously (Guttery et al, 2012) that MAP2 is involved in axoneme function, and impaired axoneme movement may prevent exflagellation. The MTOC separation that is also affected in the *map2* mutant may be another reason why gametes are unable to undergo exflagellation. The phenotype resembles that of other mutants during male gametogenesis like *crk5*, *kinesin-8B*, *kinesin-13*, and *cdc20* gene knockouts (Guttery et al, 2012; Zeeshan et al, 2019, 2022b; Balestra et al, 2020). More recently, we also observed similar chromosome segregation defects during male gametogenesis, like those seen in *cdpk4*- and *srpk1*-knockout lines, in a *nek1* knockdown line (Zeeshan et al, 2024). This suggests that these mitotic kinases drive MTOC organisation and chromosome

**Figure 4. Ultrastructure analysis of WT and *cdpk4*, *srpk1*, and *map2* gene–knockout male gametocytes.**
**(A, B)** Expansion microscopy images of WT and *cdpk4*-, *srpk1*-, and *map2*-knockout male gametocytes at 8 min post-activation. **(A)** Maximum intensity projections of whole-cell z-stack images labelled with NHS-ester (grey), α-tubulin (green), and Hoechst (blue). A magenta arrowhead indicates one of the MTOCs (WT, CDPK4-KO, and MAP2-KO) or clustered basal bodies (SRPK1-KO). Scale bar = 5 μm. **(A, B)** Single z-stack image focusing on the MTOC (WT, CDPK4-KO, and MAP2-KO) or clustered basal body (SRPK1-KO) region highlighted with a magenta arrowhead in (A), displaying NHS-ester (grey), α-tubulin (green), and Hoechst (DNA, blue). N, nucleus; BB, basal body; IM, inner spindle MTOC; OM, outer basal body MTOC. Scale bar = 2 μm. **(C)** Transmission electron microscopy images of WT and *cdpk4*-, *srpk1*-, and *map2*-knockout male gametocytes at 8 min (a, b, c, e, f, g, i, j, k, m, n, o) or 15 min (d, h, i, p) post-activation. (b, f, j, n) are magnified views of the areas enclosed by the magenta squares in (a, e, i, m), respectively. N, nucleus; K, kinetochore; A, axoneme; BB, basal body; IM, inner spindle MTOC; Mt, microtubule.

segregation. It will be interesting to study the three-dimensional organisation of these mutants during male gametogenesis using 3D electron microscopy such as serial block-face scanning electron microscopy (Hair et al, 2023).

In conclusion, this study focused on three divergent protein kinases, CDPK4, SRPK1, and MAP2, defined as male-specific because their deletion affects only male gametogenesis. The principal steps in male gametogenesis and the consequence of *cdpk4*, *srpk1*, and *map2* deletion are summarised in Fig 5. We identified their location during schizogony and male gametogenesis by fluorescence live-cell imaging, and used immunoprecipitation and mass spectrometry to identify binding partners that are potential substrates. Using expansion and electron microscopy, we detailed the ultrastructural defects during male gametogenesis in cells for which each of the genes has been deleted. Our findings provide new insights into the location and function of these PKs and their interacting partners during male gametogenesis, which may underpin the development of drugs targeting this critical stage of the *Plasmodium* life cycle.

# Materials and Methods

### Ethics statement

The animal work passed an ethical review process and was approved by the United Kingdom Home Office. Work was carried out under UK Home Office Project Licenses (30/3248 and PDD2D5182) in accordance with the UK "Animals (Scientific Procedures) Act 1986." 6- to 8-wk-old female CD1 outbred mice from Charles River Laboratories were used for all experiments.

### Generation of transgenic parasites and genotype analyses

Deletion mutants for *cdpk4*, *srpk1*, and *map2* were generated previously (Tewari et al, 2010). To generate the GFP-tag lines, a region of each gene downstream of the ATG start codon was amplified, ligated to a p277 vector, linearised using *ClaI*, and transfected as described previously (Guttery et al, 2012). The p277 vector contains the human *dhfr* cassette, conveying resistance to pyrimethamine. A schematic representation of the endogenous gene locus, the constructs, and the recombined gene locus can be found in Fig S1A, with primer sequences given in Table S1. For the parasites expressing the C-terminal GFP-tagged protein, diagnostic PCR was used with primer 1 (Int primer) and primer 3 (ol492) to confirm integration of the GFP-targeting construct (Fig S1B). WT-GFP is *P. berghei* ANKA line 507cl1 that has the *gfp* sequence stably integrated into the non-essential *230p* gene locus to constitutively express GFP under the control of the *eukaryotic elongation factor 1A* (*eef1aa*) promoter (Janse et al, 2006).

### Live-cell imaging

To examine CDPK4-GFP, SRPK1-GFP, and MAP2-GFP expression during erythrocytic stages, parasites growing in schizont culture medium were used for imaging at different stages of schizogony. Purified gametocytes were examined for the timing and location of GFP expression at different time points (0, 30 s, to 15 min) after activation in ookinete culture medium (Menard, 2013; Zeeshan et al, 2019). Images were captured using a 63x oil immersion objective on a Zeiss Axio Imager M2 microscope fitted with an AxioCam ICc1 digital camera.

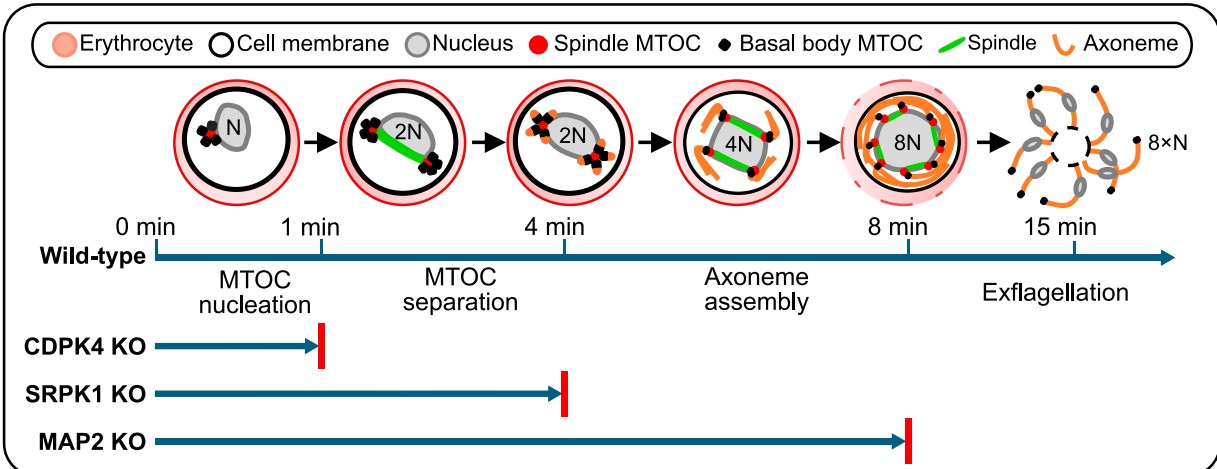

**Figure 5.  Schematic diagram of the principal steps in male gametogenesis and the consequence of *cdpk4*, s*rpk1*, and *map2* deletion.**
The top diagram illustrates *Plasmodium* male gametogenesis with the arrow below showing the main events over time in WT parasites. The activated gametocyte is haploid (N). MTOC nucleation occurs in approximately the first 1–2 min after activation. Between 2 and 4 min after activation, the first round of DNA replication (2N: diploid) takes place, the first MTOC replication and separation occurs, and axoneme synthesis begins. Within 4–6 min, the second round of DNA replication (4N: tetraploid) and MTOC replication and separation occur, and axoneme assembly continues. By 8 min post-activation, the third round of DNA replication (8N: octoploid) and MTOC replication and separation occur and axoneme synthesis is completed. Exflagellation begins, and by 15 min post-activation, 8 flagellated gametes (N) bud off from the cell body. The lower arrows indicate the progression of gametogenesis in three different protein kinase gene–knockout parasites, with the red bars indicating the proposed blocking points. In CDPK4-KO parasites, only MTOC nucleation occurs, whereas in SRPK1-KO parasites, complete axoneme assembly is blocked. In MAP2-KO parasites, axoneme assembly is completed, but exflagellation is blocked.

## Generation of dual-tagged parasite lines

Parasite lines expressing the GFP-tagged CDPK4, SRPK1, and MAP2 were mixed with a line expressing mCherry-tagged kinetochore marker, NDC80 (Zeeshan et al, 2022b), or axoneme marker, Kinesin-8B (Zeeshan et al, 2019), in equal numbers and injected into mice. Mosquitoes were fed on these mice 4–5 d after infection when gametocytaemia was high, and then examined for oocyst development and sporozoite formation at days 14 and 21 after feeding. Infected mosquitoes were allowed to feed on naïve mice, and after 4–5 d, the mice were examined for blood-stage parasitaemia by microscopy with Giemsa-stained blood smears. Gametocytes were purified, and fluorescence microscopy images were collected. Some parasites expressed both protein kinase-GFP and NDC80-mCherry; or protein kinase-GFP and Kinesin-8B-mCherry.

## Purification of gametocytes

Parasite lines were injected into phenylhydrazine-treated mice (Beetsma et al, 1998), and gametocytes were enriched by sulphadiazine treatment after 2 d of infection. The blood was collected on day 4 after infection, and gametocyte-infected cells were purified on a 48% vol/vol Nycodenz (in PBS) gradient (Nycodenz stock solution: 27.6% wt/vol Nycodenz in 5 mM Tris–HCl, pH 7.20, 3 mM KCl, 0.3 mM EDTA). The gametocytes were harvested from the interface and activated as described previously (Zeeshan et al, 2020).

## Immunoprecipitation and mass spectrometry

Male gametocytes of CDPK4-GFP, SRPK1-GFP, and MAP2-GFP lines were used at 6 min post-activation to prepare cell lysates. WT-GFP gametocytes were used as controls. Purified parasite pellets were cross-linked using formaldehyde (10-min incubation with 1% formaldehyde, followed by 5-min incubation in 0.125 M glycine solution and three washes with PBS [pH 7.5]). The cross-linked samples were solubilised in lysis buffer (10 mM Tris–HCl, pH 7.5, 150 mM NaCl, 0.5 mM EDTA, 0.5% Nonidet P-40 Substitute, 0.09% sodium azide with 1 x cOmplete, EDTA-free protease inhibitor cocktail [Roche]) with sonication for 1 min and lysis for 30 min on ice. Immunoprecipitation was performed using the protein lysates and GFP-Trap_A Kit (ChromoTek) following the manufacturer's instructions. Briefly, the lysates were incubated for 2 h with GFP-Trap_A beads at 4°C with continuous rotation. Unbound proteins were washed away, and proteins bound to the GFP-Trap_A beads were digested using trypsin. The tryptic peptides were analysed by liquid chromatography–tandem mass spectrometry. Mascot (http://www.matrixscience.com/) and MaxQuant (https://www.maxquant.org/) search engines were used for mass spectrometry data analysis. Peptides and proteins having a minimum threshold of 95% were used for further proteomics analysis. The PlasmoDB database was used for protein annotation (Aurrecoechea et al, 2009), and Gene Ontology analysis was performed using the tool available through PlasmoDB (http://PlasmoDB.org) with the following settings: computed and curated evidence allowed, use GO Slim terms, and P-value cut-off = 0.05.

## Ultrastructure expansion microscopy

Sample preparation for U-ExM was performed based on previously described protocols (Rashpa & Brochet, 2022; Liffner et al, 2024). Purified and 8-min activated gametocytes were fixed in 4% formaldehyde in PHEM buffer (60 mM Pipes, 25 mM Hepes, 10 mM EGTA, 2 mM MgCl$_2$, pH 6.9) at room temperature for 15 min. Fixed samples were attached to 10-mm round poly-D-lysine–coated coverslips for 15 min. Coverslips were incubated overnight at 4°C in 1.4% formaldehyde (FA)/2% acrylamide (AA). Gelation was performed in ammonium persulphate/TEMED (10% each)/monomer solution (23% sodium acrylate; 10% AA; 0.1% BIS-AA in PBS) on ice for 5 min and at 37°C for 30 min. Gels were denatured for 15 min at 37°C and for 45 min at 95°C in denaturation buffer (200 mM SDS, 200 mM NaCl, 50 mM Tris, pH 9.0, in water). After denaturation, gels were incubated in distilled water overnight for complete expansion. The next day, circular gel pieces with a diameter of ~13 mm were excised, and the gels were washed in PBS three times for 15 min to remove excess water. The gels were then incubated in blocking buffer (3% BSA in PBS) at room temperature for 30 min, incubated with mouse monoclonal anti-α-tubulin antibody (T9026; Sigma-Aldrich) in blocking buffer (1:500 dilution) at 4°C overnight, and washed three times for 15 min in wash buffer (0.5% vol/vol Tween-20 in PBS). The gels were incubated with 8 μg/ml Atto 594 NHS-ester (Merck), 10 μg/ml Hoechst 33342 (Molecular Probes), and Alexa Fluor 488 goat anti-mouse IgG (A11001; Invitrogen) in PBS (1:500 dilution) at 37°C for 3 h followed by three washes of 15 min each in wash buffer (blocking and all antibody incubation steps were performed with gentle shaking). The gels were then washed three times for 15 min with wash buffer and expanded overnight in ultrapure water. The expanded gel was placed in a 35-mm glass-bottom dish (MatTek) with the 14-mm glass coated with poly-D-lysine and mounted with an 18 × 18 mm coverslip to prevent the gel from sliding and to avoid drifting while imaging. High-resolution confocal microscopy images were acquired using a Zeiss Celldiscoverer 7 with Airyscan using a Plan-Apochromat 50×/1.2 NA Water objective, with 405-, 488-, and 561-nm lasers. Confocal z-stacks were acquired using line scanning and the following settings: 55 × 55 nm pixel size, 170-nm z-step, 2.91-μs/pixel dwell time, 850 gain, and 3.5% (405 nm), 4.5% (488 nm), and 5.0% (561 nm) laser powers. The z-stack images were processed and analysed using Fiji (version 1.54f; Schindelin et al, 2012).

## Electron microscopy

Gametocytes activated for 8 and 15 min were fixed in 4% glutaraldehyde in 0.1 M phosphate buffer and processed for electron microscopy (Ferguson et al, 2005). Briefly, samples were post-fixed in osmium tetroxide, treated en bloc with uranyl acetate, dehydrated, and embedded in Spurr's epoxy resin. Thin sections were stained with uranyl acetate and lead citrate before examination in a Tecnai G2 12 BioTwin (FEI UK, UK) or a JEOL 1200EX electron microscope (JEOL Ltd).

# Data Availability

The mass spectrometry proteomics data from this publication have been deposited to the ProteomeXchange Consortium via the PRIDE

(Perez-Riverol et al, 2025; https://www.ebi.ac.uk/pride/) partner repository with the dataset identifier PXD061651.

## Supplementary Information

## Acknowledgements

We wish to thank Julie Rodgers for helping to maintain the insectary and other technical work and Dr Benoit Poulin for assistance. We thank the Nanoscale and Microscale Research Centre (nmRC) for providing access to instrumentation and Dr Michael W Fey, Dr Julie Watts, and Ms Nicola J Weston for technical assistance. This work was supported by MRC UK (G0900109, G0900278, MR/K011782/1) to R Tewari, BBSRC (BB/N017609/1) and ERC advance grant funded by UKRI Frontier Science (EP/X024776/1) to R Tewari, BBSRC (BB/X014681/1) to DS Guttery and R Tewari, and BBSRC (BB/N017609/1). R Yanase is supported under grant BB/X014681/1, and M Zeeshan is supported through grants EP/X024776/1 and BB/N017609/1. AA Holder is supported by the Francis Crick Institute (FC001097), which receives its core funding from Cancer Research UK (FC001097), the UK Medical Research Council (FC001097), and the Wellcome Trust (FC001097). For Open Access, the authors have applied a CC BY public copyright licence to any Author Accepted Manuscript version arising from this submission.

### Author Contributions

R Yanase: data curation, formal analysis, validation, investigation, visualisation, methodology, and writing—original draft, review, and editing.

M Zeeshan: data curation, formal analysis, validation, investigation, visualisation, methodology, and writing—review and editing.

DJP Ferguson: resources, data curation, formal analysis, validation, investigation, visualisation, methodology, and writing—review and editing.

R Markus: visualisation and methodology.

D Brady: formal analysis, validation, investigation, and methodology.

AR Bottrill: resources, data curation, and methodology.

AA Holder: formal analysis, supervision, funding acquisition, validation, methodology, project administration, and writing—original draft, review, and editing.

DS Guttery: supervision, funding acquisition, validation, visualisation, project administration, and writing—original draft, review, and editing.

R Tewari: conceptualisation, resources, formal analysis, supervision, funding acquisition, validation, investigation, visualisation, methodology, project administration, and writing—original draft, review, and editing.

### Conflict of Interest Statement

The authors declare that they have no conflict of interest.

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
