## [Reviewer comments · Life Science Alliance]

Life Science Alliance

Divergent Plasmodium kinases drive MTOC, kinetochore and axoneme organisation in male gametogenesis

Ryuji Yanase, Mohammad Zeeshan, David Ferguson, Robert Markus, Declan Brady, Andrew Bottrill, Anthony Holder, David Guttery, and Rita Tewari

DOI: <https://doi.org/10.26508/lsa.202403056>

Corresponding author(s): Rita Tewari, University of Nottingham and David Guttery, UNIVERSITY OF LEICESTER

Review Timeline:

Submission Date:	2024-09-22
Editorial Decision:	2024-11-06
Revision Received:	2025-03-04
Editorial Decision:	2025-03-06
Revision Received:	2025-03-10
Accepted:	2025-03-11

Transaction Report:

November 6, 2024

Re: Life Science Alliance manuscript #LSA-2024-03056

Prof. Rita Tewari
University of Nottingham
School of Life Sciences
Queens Medical Centre
Nottingham NG7 2UH
United Kingdom

Dear Dr. Tewari,

Thank you for submitting your manuscript entitled "Divergent Plasmodium protein kinases drive MTOC, kinetochore and axoneme organisation in male gametogenesis" to Life Science Alliance. The manuscript was assessed by expert reviewers, whose comments are appended to this letter. We invite you to submit a revised manuscript addressing the Reviewer comments.

Thank you for this interesting contribution to Life Science Alliance. We are looking forward to receiving your revised manuscript.

Sincerely,

B. MANUSCRIPT ORGANIZATION AND FORMATTING:

Reviewer #1 (Comments to the Authors (Required)):

Yanase et al. examine the roles of three male-specific protein kinases (PKs), namely calcium-dependent protein kinase-4 (CDPK4), serine/arginine-rich protein kinase (SRPK1) and mitogen-activated protein kinase-2 (MAP2), in male gametocyte activation in *P. berghei*.

Some points need to be addressed.

It may be useful to provide additional clarity around the term, MTOC. As the authors point out, MTOCs have two main functions: the organization of the mitotic and meiotic spindle apparatus (centrosome or centrosome equivalent) and the organization of eukaryotic flagella and cilia (basal bodies).

When applied to male plasmodium gametocytes the terminology gets a little confusing. For example, the following statement could cause confusion among some readers. "Throughout DNA replication the MTOC is crucial to its completion... The MTOC has different names in different species; for example, in yeast it is known as the spindle pole body (SPB), in humans, the centrosome, and in ciliated cells, the basal body."

The basal body is not involved in DNA replication. It may be best to refer to the two different MTOCs as the spindle MTOC and the basal body MTOC. And to define how this relates to the term, axoneme. It would also be useful to expand the description of the bipartite MTOC of male gametocytes. Does the bipartite MTOC comprise the spindle MTOC and the Basal Body, and thus play roles in both DNA division and axoneme formation?

The authors initially used live-cell imaging to examine the locations of the three PKs, during the rapid male gamete formation process. They claim that PbCDPK4 and PbSRPK1 are diffusely distributed throughout the parasite cell, whereas PbMAP2 is only in the nucleus and only during male gametogenesis. In fact, the data suggest that PbSRPK is largely excluded from the nucleus while PbMAP2 is found only in the nucleus. Nonetheless the authors later suggest that the three PKs interact with each other. How is this possible?

Figure S1B. What is WT-GFP? Is it parasites transfected to express unfused GFP? If so, please include a description in the Methods? Fluorescence microscopy images of this control would provide a useful control, as unfused GFP is likely distributed evenly across the cytoplasm and the nucleus).

Figure S1B. Please use arrows to point to the bands of interest. The SRPK is very faint. Does this correlate with the level of GFP expression? Full-length blots should be provided in a separate Supp Figure.

The value of Figure S1C is not clear. There is a need for additional examples of the different microscopy images. But these images of activated gametocytes are difficult to interpret. How are these different from the data shown in Fig 1B? At least show the different panels for the staining depicted in blue (presumably, this is Hoechst, though it is not stated in the legend).

Line 88. "SRPK1-GFP showed weak diffuse cytoplasmic expression with a strong focus at the location of the nuclear pole and towards the apex of the forming or mature merozoites". I agree that the labelling is towards the apex of the forming or mature merozoites, but in some cases quite distant from the nuclear membrane. It would be interesting to look for co-location with a centrin marker in this stage.

Line 110. "At the early stage of male gametogenesis (30 s to 6 min post-activation), up to two foci of CDPK4-GFP were observed, whereas at the late stage (>6 min post-activation, more than three foci were sometimes observed". It would be useful to provide an example of a cell with 3 foci.

Line 113. "SRPK1-GFP was located in the cytoplasm, concentrically surrounding the nucleus." It is not clear what is meant by "concentrically surrounding the nucleus." Do the authors mean that it is largely excluded from the nucleus?

Line 114. "MAP2-GFP had a nuclear location in activated male gametocytes (Figure 1B)." The images reveal that MAP2-GFP is present in both the cytoplasm and the nucleus. The statement could say that "MAP2-GFP is preferentially located in the nucleus in activated male gametocytes".

The authors suggest that the SRPK1-GFP-labelled structure is surrounded by kinesin-8B-mCh. However, these proteins appear to show little co-localisation. Thus, the current data do not support the suggestion that SRPK1 interacts with axoneme proteins or proteins involved in their assembly. Analysis of the level of overlap at a pixel-by-pixel level in confocal sections through the cell could provide further evidence for the proposed overlap.

The authors used anti-GFP antibody to immunoprecipitate CDPK4-GFP, SRPK1-GFP and MAP2-GFP from lysates of 6 min post-activated gametocytes, identifying 108, 201 and 162 proteins. While this is useful data, none of these proposed interactions has been validated; thus, it is important to be careful with the interpretation. For example, the immunofluorescence experiment suggests that SRPK1-GFP is largely excluded from the nucleus. Yet the immunoprecipitated proteins include mini-chromosome maintenance (MCM) proteins which should be nuclear located.

The authors should also discuss the underlying assumption that there should be a physical association of the kinases with proposed targets that can be assessed by immunoprecipitation. Phosphorylation of targets by the kinases might be expected to take place during a very transient association. Similarly, kinases may be components of a complex or compartment that is immunoprecipitated, but this does not necessarily mean that all of these proteins are targets.

The authors then determine the morphological consequences of the knockout of CDPK4, SRPK1 and MAP2 using ultrastructure expansion microscopy (UEXM) and transmission electron microscopy (TEM).

The amount of data provided is limited which makes the images a little hard to interpret. For example, the higher magnification TEM panels are not zoom-in versions of the left hand panels. It would be useful to provide additional examples of both the UEXM and TEM data in the Supp Info.

For the WT male gametes, it is not clear why the kinetochores are distributed along the spindle in (b) but contracted to a point below the MTOC in (a). The basal body is not evident in (a), so it is not clear if this is developing from, or associated with, the structure labelled IM.

Each of the knock-outs has an interesting phenotype. Additional images of the cells would facilitate analysis of each of the phenotypes.

In the CDPK4 knockout parasites, the axonemes fail to form. In the single UEXM CDPK4 knockout parasite image, the magenta arrow points to a protein-dense structure. What is the evidence that this is an MTOC? The data suggest loss of the nexus between the spindle and basal body. The authors suggest the phenotype is reminiscent of that of a kinesin-13 deficient line, consistent with their finding that CDPK4 may interact with kinesin-13. However, molecular basis of the phenotype is likely much more complex than lack of phosphorylation of a single target protein. The phenotype is also consistent with knock-out/knock-down studies of several kinetochore (and associated) proteins (Farrell and Gubbels, 2014) (Brusini et al., 2022) (Li et al., 2024). The authors should discuss more fully.

In the SRPK1 knockout, the axonemes form, but fail to segregate correctly. Again, there appears to be a disruption of the spindle/ spindle MTOC/ BB nexus. It would be useful if this separation of kinetochores from the nucleus periphery could be quantitated.

In the MAP2 knockout, the phenotype is less dramatic, and the nexus between the spindle/ spindle MTOC/ BB nexus is apparently maintained, but duplicated spindle MTOCs appear to fail to segregate. Again, quantification of the observed phenotypes would strengthen the conclusions.

Further immunofluorescence characterization, with antibodies recognizing the different spindle and MTOC/ apical components, would be rewarding. However, I appreciate that this is outside the scope of this study.

In summary, this is an interesting study of protein kinases (PKs) in male gametocyte activation in *P. berghei*. The data add to our understanding of the roles of these kinases, but care should be taken not to overinterpret the immunoprecipitation studies.

References

- Brusini, L., N. Dos Santos Pacheco, E.C. Tromer, D. Soldati-Favre, and M. Brochet. 2022. Composition and organization of kinetochores show plasticity in apicomplexan chromosome segregation. *J Cell Biol.* 221.
- Farrell, M., and M.J. Gubbels. 2014. The *Toxoplasma gondii* kinetochore is required for centrosome association with the centrocone (spindle pole). *Cell Microbiol.* 16:78-94.
- Li, J., G.J. Shami, B. Liffner, E. Cho, F. Braet, et al. 2024. Disruption of *Plasmodium falciparum* kinetochore proteins destabilises the nexus between the centrosome equivalent and the mitotic apparatus. *Nat Commun.* 15:5794.

Reviewer #2 (Comments to the Authors (Required)):

The manuscript reports findings relative to the specific localization and function of three kinases in the rodent malaria species *P. berghei*. Namely, authors explore the localization of PbCDPK4, SRPK1 and MAP2. Though mutant phenotypes previously pointed at their functions being restricted to axoneme formation, basal body positioning and chromosome condensation, respectively, their localization and how they were involved - mechanistically- in these processes was not known. Authors tackle these unexplored aspects of these kinases by live imaging and by identifying SRPK1 and MAP2 interacting partners. Mutant phenotypes are further analyzed by ultrastructure expansion microscopy.

An important distinction should be made between interacting partners and substrates. Immunoprecipitation assays can at best identify interacting partners, which may or may not be the kinases' substrates. Substrate identification would entail direct evidence, such as *in vitro* phosphorylation or changes in the phosphorylation status demonstrated for example by specific antibody recognition of a phospho version of the protein or phosphoproteomics upon, for example, kinase knock down or pharmacological inhibition. This clarification is only done in the last paragraph of the results/discussion section, but the abstract, for example, makes no distinction. In fact, it mentions that the manuscript "identifies putative substrates" which it does not. Just to name another example, Line 119 specifies that the results are compatible with "a role of reversible protein phosphorylation as a key driver of male gametogenesis" However, phosphorylation is not directly addressed in this work.

The concept of "divergent" necessarily entails a comparison - divergent in terms of? When compared to? The term is used throughout the manuscript as an adjective describing the kinases as if it were accepted that they are different to something, only the something is never specified. However, they are named after widely conserved kinases involved in the regulation of the same widely conserved cell division events. The authors may want to revisit this, clarifying if they chose to keep the terminology, why they are coined "divergent kinases".

It is conspicuous that all interacting partners of CDPK4 identified at the time point in which immunoprecipitation experiments were pursued were components of the replisome, given that CDPK4 is not primarily localized at the nucleus at that point. Instead, it is shown in Figure 1B to be localized to the cell poles. Are any of these interacting partners known to localize outside of the nucleus post-activation of gametocytes? Again, the authors discuss how CDPK4 might regulate its interacting partners by means of phosphorylation, however, direct phosphorylation of these putative substrates was not shown. They in fact mention that their data is in partial agreement with phosphoproteomic data previously obtained by other authors. Is it plausible that physical interactions with additional partners may play additional roles which haven't been discussed?

Overall, the manuscript would benefit from a schematic summarizing all findings. It would be useful for the reader to get a general perspective of when and where these kinases are involved in the male gamete formation process, who they interact with and what evidence is gathered from other studies exploring phospho-changes in potential substrates as well as interacting partners. The model can also summarize how the authors envision these kinases "talk to each other" through the process.

Minor points

Line 111 refers to the analysis of CDPK4 localization at >6 min time points, however, this is not shown (or where it is shown is not immediate neither is it specified).

Line 114 specified that MAP2-GFP displays a nuclear localization. While this is clearly the case, it is also detectable in the cytosol, albeit not as markedly.

Reviewer #3 (Comments to the Authors (Required)):

Summary

In this manuscript 3 previously studied kinases are revisited in a comparative fashion to flesh out where they differ in the endomitotic steps during male gametocytogenesis. This is important as timing of analysis is super-critical in this sec-min timescale process, which makes it hard to comparatively interpret studies from different laboratories, all using their own, different protocols. Besides the comparative angle, new insights are the pull-down proteomes for all 3 kinases, which partially overlap with previous phosphoproteomes - but this is a different level of data, which could be emphasizes more in the manuscript. Furthermore, expansion microscopy (U-ExM) is applied next to classic TEM. Overall, non-overlapping functions for the kinases are described as well as tantalizing hints as how chromosome condensation might be working in this process. Although the title suggests a focus on the MTOC, kinetochore and axoneme, the scope is beyond that by uncovering roles in ORCs, MCMs, and SMCs.

Specific points

My only pause is describing the pulled-down proteins with each kinase as substrates. They might be substrates but might be proteins in complex with substrates, or structural proteins used for spatial cues of the kinases. This should be interpreted more carefully.

Response to reviewers

We thank the reviewers for their valuable suggestions and comments. In response to these comments, we have conducted additional experiments and revised the manuscript. Below, highlighted in blue, are our point-by-point responses.

*Line numbers correspond to those in the track-changes file.

Reviewer #1:

Yanase et al. examine the roles of three male-specific protein kinases (PKs), namely calcium-dependent protein kinase-4 (CDPK4), serine/arginine-rich protein kinase (SRPK1) and mitogen-activated protein kinase-2 (MAP2), in male gametocyte activation in *P. berghei*.

Some points need to be addressed.

1. It may be useful to provide additional clarity around the term, MTOC. As the authors point out, MTOCs have two main functions: the organization of the mitotic and meiotic spindle apparatus (centrosome or centrosome equivalent) and the organization of eukaryotic flagella and cilia (basal bodies).

As we and others have shown, during male gametogenesis *Plasmodium* spp. elaborate a bipartite MTOC (Rashpa & Brochet, 2022; Zeeshan et al, 2022a). Gametogenesis is a mitotic cell division, and the resultant male gamete is the only flagellated stage of the life cycle. Immediately after the activation of the male gametocyte, a paired bipartite MTOC spanning the nuclear membrane is formed, consisting of an inner acentriolar MTOC and an outer cytoplasmic centriolar MTOC. During mitosis, the inner acentriolar MTOC (also known as the nuclear pole) organises the mitotic spindle microtubules in the nucleus. The outer cytoplasmic centriolar MTOC assembles a basal body and an axoneme for flagellum synthesis. These two regions of the bipartite MTOC are coordinated to ensure that each gamete has one nucleus and one flagellum. To make a clear distinction between these two MTOCs, we now refer to the inner acentriolar MTOC as the 'spindle MTOC' and the outer cytoplasmic centriolar MTOC as the 'basal body MTOC', as suggested by the reviewer (Lines 65-71).

2. When applied to male plasmodium gametocytes the terminology gets a little confusing. For example, the following statement could cause confusion among some readers. "Throughout DNA replication the MTOC is crucial to its completion... The MTOC has different names in different species; for example, in yeast it is known as the spindle pole body (SPB), in humans, the centrosome, and in ciliated cells, the basal body."

We have revised the MTOC description in the manuscript using the terms 'spindle MTOC' and 'basal body MTOC' to clarify the distinction between the outer and inner MTOCs in *Plasmodium* male gametocytes (Lines 62-71).

3. The basal body is not involved in DNA replication. It may be best to refer to the two different MTOCs as the spindle MTOC and the basal body MTOC. And to define how this relates to the term, axoneme. It would also be useful to expand the description of the bipartite MTOC of male gametocytes. Does the bipartite MTOC comprise the spindle MTOC and the Basal Body, and thus play roles in both DNA division and axoneme formation?

As mentioned above, we use ‘spindle MTOC’ and ‘basal body MTOC’ in the revised manuscript, following the reviewer’s suggestion. We have also expanded the description of the bipartite MTOC to clarify the role of each MTOC (Lines 62-73).

4. The authors initially used live-cell imaging to examine the locations of the three PKs, during the rapid male gamete formation process. They claim that PbCDPK4 and PbSRPK1 are diffusely distributed throughout the parasite cell, whereas PbMAP2 is only in the nucleus and only during male gametogenesis. In fact, the data suggest that PbSRPK is largely excluded from the nucleus while PbMAP2 is found only in the nucleus. Nonetheless the authors later suggest that the three PKs interact with each other. How is this possible?

We agree, as the reviewer suggests in comment 11, that PbMAP2 is present in both the cytoplasm and the nucleus and is preferentially located in the nucleus in activated male gametocytes (Lines 219-220). We detected no interaction between SRPK1 and MAP2 in the pulldown data, but detected a weak interaction between CDPK4 and MAP2 (Table 1), which may indicate an interaction between these PKs in the cytoplasm.

5. Figure S1B. What is WT-GFP? Is it parasites transfected to express unfused GFP? If so, please include a description in the Methods? Fluorescence microscopy images of this control would provide a useful control, as unfused GFP is likely distributed evenly across the cytoplasm and the nucleus).

WT-GFP is *Plasmodium berghei* ANKA 507c11 line that has *gfp* stably integrated into the non-essential *230p* gene locus to express GFP under the control of the constitutive *eukaryotic elongation factor 1A (eef1aa)* promoter (Janse et al, 2006). This line expresses GFP without any drug-resistance markers, which helps in the visualization and counting of parasites. This description has now been added to the Materials and Methods (Generation of transgenic parasites and genotype analysis). In addition, we include in Figure S1C fluorescence microscopy images of these control cells expressing WT-GFP. As the reviewer suggests, unfused GFP is distributed evenly throughout the cell.

6. Figure S1B. Please use arrows to point to the bands of interest. The SRPK is very faint. Does this correlate with the level of GFP expression? Full-length blots should be provided in a separate Supp Figure.

We have added arrows to point to the bands of interest in Figure S1B (Molecular masses: 87.6 kDa [CDPK4-GFP], 180.7 kDa [SRPK1-GFP], and 88.1 kDa [MAP2-GFP]). These western blots were performed to confirm the expression of GFP-tagged PKs at the correct size and without degradation. The number of parasites used for each western blot was not matched precisely and for a variety of reasons there is not an appropriate internal loading control, therefore the band intensity on each western blot is not expected to reflect the relative expression levels of the GFP-fused PKs. We have provided uncropped full-length blots and integration PCR gel images in a separate supplementary data (SourceData.pdf).

7. The value of Figure S1C is not clear. There is a need for additional examples of the different microscopy images. But these images of activated gametocytes are difficult to interpret. How

are these different from the data shown in Fig 1B? At least show the different panels for the staining depicted in blue (presumably, this is Hoechst, though it is not stated in the legend).

We have improved the images, with additional examples of images. We include fluorescence microscopy images of WT-GFP control cells in Figure S1C so that the readers can more readily distinguish the different patterns of expression of the GFP-tagged PKs in male gametocytes. We also provide different panels for Hoechst and DIC in Figure S1C.

8. Line 88. "SRPK1-GFP showed weak diffuse cytoplasmic expression with a strong focus at the location of the nuclear pole and towards the apex of the forming or mature merozoites". I agree that the labelling is towards the apex of the forming or mature merozoites, but in some cases quite distant from the nuclear membrane. It would be interesting to look for co-location with a centrin marker in this stage.

We performed immunofluorescence microscopy using asexual blood stages of the SRPK1-GFP cell line with anti-GFP and anti-centrin or anti- α -tubulin antibodies. SRPK1 was located at a position away from that of centrin, and near the tip of the microtubules extending toward the apex of the merozoite. Figure S2B shows these results.

9. Line 110. "At the early stage of male gametogenesis (30 s to 6 min post-activation), up to two foci of CDPK4-GFP were observed, whereas at the late stage (>6 min post-activation, more than three foci were sometimes observed)". It would be useful to provide an example of a cell with 3 foci.

Images of male gametocytes at the late stage, which have more than three CDPK4-GFP foci, are now in Figure S1D.

10. Line 113. "SRPK1-GFP was located in the cytoplasm, concentrically surrounding the nucleus." It is not clear what is meant by "concentrically surrounding the nucleus." Do the authors mean that it is largely excluded from the nucleus?

The reviewer is correct, we meant 'it is largely excluded from the nucleus'. We have revised this part of the manuscript as suggested by the reviewer (Line 218).

11. Line 114. "MAP2-GFP had a nuclear location in activated male gametocytes (Figure 1B)." The images reveal that MAP2-GFP is present in both the cytoplasm and the nucleus. The statement could say that "MAP2-GFP is preferentially located in the nucleus in activated male gametocytes".

We agree with the reviewer's suggestion that MAP2-GFP is present in both cytoplasm and the nucleus, but preferentially located in the nucleus in activated gametocytes. We have revised this part of the manuscript (Lines 219-220).

12. The authors suggest that the SRPK1-GFP-labelled structure is surrounded by kinesin-8B-mCh. However, these proteins appear to show little co-localisation. Thus, the current data do not support the suggestion that SRPK1 interacts with axoneme proteins or proteins involved in their assembly. Analysis of the level of overlap at a pixel-by-pixel level in confocal sections through the cell could provide further evidence for the proposed overlap.

We thank the reviewer for this observation. The co-localisation of SRPK1-GFP and kinesin-8B-mCh is not strong, and it was an over-interpretation to suggest that these data strongly support an interaction between SRPK1 and axoneme proteins. After carefully examination of other co-localisation data for SRPK1-GFP and kinesin-8B-mCh at a pixel-by-pixel level, we identified some areas of co-localisation and other areas of essentially none. These data are shown in Figure S3, and we have revised the relevant text (Lines 229-230).

13. The authors used anti-GFP antibody to immunoprecipitate CDPK4-GFP, SRPK1-GFP and MAP2-GFP from lysates of 6 min post-activated gametocytes, identifying 108, 201 and 162 proteins. While this is useful data, none of these proposed interactions has been validated; thus, it is important to be careful with the interpretation. For example, the immunofluorescence

experiment suggests that SRPK1-GFP is largely excluded from the nucleus. Yet the immunoprecipitated proteins include mini-chromosome maintenance (MCM) proteins which should be nuclear located.

The reviewer is correct to be cautious about the interpretation of these data. Whilst an interaction between SRPK1-GFP and MCM proteins may seem unlikely, it has been suggested that MCM proteins can be found in both nucleus and cytoplasm, where they interact with kinases and phosphatases (Abe et al, 2012). It is possible that the apparent interaction between SRPK1-GFP and MCM proteins detected by immunoprecipitation may reflect the presence of both in the cytoplasm. We have reviewed our description of the immunoprecipitation data and revised it as appropriate (Lines 283-288).

14. The authors then determine the morphological consequences of the knockout of CDPK4, SRPK1 and MAP2 using ultrastructure expansion microscopy (UExM) and transmission electron microscopy (TEM).

The amount of data provided is limited which make the images a little hard to interpret. For example, the higher magnification TEM panels are not zoom-in versions of the left hand panels. It would be useful to provide additional examples of both the UExM and TEM data in the Supp Info.

In Figure 4C, panels b, f, j and n are zooms of the regions highlighted in panels a, e, i and m and this has been clarified in the Figure 4 legend. We include additional U-ExM and TEM data in Figure S4 and S5.

15. For the WT male gams, it is not clear why the kinetochores are distributed along the spindle in (b) but contracted to a point below the MTOC in (a). The basal body is not evident in (a), so it is not clear if this is developing from, or associated with, the structure labelled IM.

We apologise for the confusion around these TEM images, and for the discrepancy between the overall view of the cell and the magnified view of the MTOC, which may have caused misunderstanding. The images in Figure 4C have been replaced so that there is consistency between the overall view of the cell and the magnified view of the MTOC. These TEM images show kinetochores captured by microtubules extending from the spindle MTOC and contracting toward this inner spindle MTOC.

16. Each of the knock-outs has an interesting phenotype. Additional images of the cells would facilitate analysis of each of the phenotypes.

Additional U-ExM and TEM images of the gene knock-out parasite lines are included in Figures S4 and S5.

17. In the CDPK4 knockout parasites, the axonemes fail to form. In the single UExM CDPK4 knockout parasite image, the magenta arrow points to a protein-dense structure. What is the evidence that this is an MTOC? The data suggest loss of the nexus between the spindle and basal body. The authors suggest the phenotype is reminiscent of that of a kinesin-13 deficient line, consistent with their finding that CDPK4 may interacts with kinesin-13. However, molecular basis of the the phenotype is likely much more complex than lack of phosphorylation

of a single target protein. The phenotype is also consistent with knock-out/ knock-down studies of several kinetochore (and associated) proteins (Farrell and Gubbels, 2014) (Brusini et al., 2022) (Li et al., 2024). The authors should discuss more fully.

The single protein-dense structure in the U-ExM image of a CDPK4-KO cell is a clear structure across the nuclear membrane, which is a distinctive feature of the bipartite MTOC of male gametocytes. A similar single bipartite MTOC was observed by U-ExM of CDPK4-KO male gametocytes in a previous study (Rashpa & Brochet, 2022), and centrin was shown to localise to this MTOC. In addition, a short microtubular structure, considered to be a precursor of an axoneme that had failed to form, is extending from the outer protein-dense structure. We consider this protein-dense structure to be a basal body MTOC that failed to duplicate and separate. We agree with the reviewer's comment that 'the molecular basis of the phenotype is likely much more complex than lack of a single target protein', and we have revised the discussion of the gene knockout phenotypes, including reference to additional literature (Lines 373-375 and 382-399) (Fang et al, 2017; Invergo et al, 2017; Kumar et al, 2022).

18. In the SRPK1 knockout, the axonemes form, but fail to segregate correctly. Again, there appears to be a disruption of the spindle/ spindle MTOC/BB nexus. It would be useful if this separation of kinetochores from the nucleus periphery could be quantitated.

We examined the TEM data for 29 kinetochores from 12 WT male gametocytes and 20 kinetochores from 10 SRPK1-KO male gametocytes. We found that all WT kinetochores (29/29) were associated with spindles, whereas no SRPK1-KO kinetochores (0/20) were associated with spindles. All SRPK1-KO kinetochores were separate from the periphery of the nucleus. These quantitative data are included in the manuscript (Line 402-405).

19. In the MAP2 knockout, the phenotype is less dramatic, and the nexus between the spindle/ spindle MTOC/ BB nexus is apparently maintained, but duplicated spindle MTOCs appear to fail to segregate. Again, quantification of the observed phenotypes would strengthen the conclusions.

We examined the U-ExM data for 28 MTOCs from 5 WT male gametocytes and 34 MTOCs from 6 MAP2-KO male gametocytes. We found that all WT MTOCs (28/28) were segregated, whereas only 38% of MAP2-KO MTOCs segregated and the remainder (21/34) failed (Lines 416-419).

20. Further immunofluorescence characterization, with antibodies recognizing the different spindle and MTOC/ apical components, would be rewarding. However, I appreciate that this is outside the scope of this study.

As indicated in our response to comment 8, we have included immunofluorescence data showing the localisation of SRPK1, with centrin or α -tubulin in Figure S2B.

21. In summary, this is an interesting study of protein kinases (PKs) in male gametocyte activation in *P. berghei*. The data add to our understanding of the roles of these kinases, but care should be taken not to overinterpret the immunoprecipitation studies.

We appreciate the reviewer's valuable comments. We have carefully considered our interpretation of the immunoprecipitation studies and revised the manuscript to stress that these proteins are potential interacting partners rather than substrates. We believe that with the additional data and revisions based on the reviewers' comments, we have improved the manuscript.

Reviewer #2:

The manuscript reports findings relative to the specific localization and function of three kinases in the rodent malaria species *P. berghei*. Namely, authors explore the localization of PbCDPK4, SRPK1 and MAP2. Though mutant phenotypes previously pointed at their functions being restricted to axoneme formation, basal body positioning and chromosome condensation, respectively, their localization and how they were involved - mechanistically- in these processes was not known. Authors tackle these unexplored aspects of these kinases by live imaging and by identifying SRPK1 and MAP2 interacting partners. Mutant phenotypes are further analyzed by ultrastructure expansion microscopy.

1. An important distinction should be made between interacting partners and substrates. Immunoprecipitation assays can at best identify interacting partners, which may or may not be the kinases' substrates. Substrate identification would entail direct evidence, such as in vitro phosphorylation or changes in the phosphorylation status demonstrated for example by specific antibody recognition of a phospho version of the protein or phosphoproteomics upon, for example, kinase knock down or pharmacological inhibition. This clarification is only done in the last paragraph of the results/discussion section, but the abstract, for example, makes no distinction. In fact, it mentions that the manuscript "identifies putative substrates" which it does not. Just to name another example, Line 119 specifies that the results are compatible with "a role of reversible protein phosphorylation as a key driver of male gametogenesis" However, phosphorylation is not directly addressed in this work.

We appreciate the reviewer's critical comment. As the reviewer points out, we have not directly identified any actual substrates or examined the phosphorylation of the three kinases. We have revised the manuscript to make clear that immunoprecipitation can only identify potential interacting partners, and this distinction between interacting partners and substrates is clarified throughout the manuscript.

2. The concept of "divergent" necessarily entails a comparison - divergent in terms of? When compared to? The term is used throughout the manuscript as an adjective describing the kinases as if it were accepted that they are different to something, only the something is never specified. However, they are named after widely conserved kinases involved in the regulation of the same widely conserved cell division events. The authors may want to revisit this, clarifying if they chose to keep the terminology, why they are coined "divergent kinases".

CDPK4, SRPK1, and MAP2 are PKs with orthologs in other eukaryotes and named after widely conserved kinases involved in the regulation of cell division in other eukaryotes. Despite the clear and significant sequence homology, their function and their substrates are thought to exhibit significant specificity in *Plasmodium* parasites. For example, these three protein kinases are known to function during male gametogenesis, a unique biological process, and have crucial roles in the rapid DNA replication and flagellated gamete formation that are

the hallmarks of male gametogenesis. It has been suggested that apicomplexan CDPKs have acquired several unique domains, likely reflecting their diverse function (Billker et al, 2009). Phylogenetic analysis of kinase domain sequences characterised *Plasmodium* MAP kinases including MAP2 as atypical and a conserved family distinct from classical MAP kinases (Tewari et al, 2005). *P. berghei* SRPK1 differs from homologs in other eukaryotes in its substrate specificity and its role in regulating transcription factors. While mammalian SRPK1 primarily phosphorylates splicing factors, *P. berghei* SRPK1 may have a broader substrate specificity and is involved in regulating the expression of genes involved in various cellular processes (Dixit et al, 2010).

‘Divergent’ refers to the significant differences in sequence, domain architecture, predicted substrate specificity and function between the *Plasmodium* kinases and their orthologs in other eukaryotes. While these kinases have functional similarities with their counterparts in regulating conserved cellular processes like cell division, their unique structural features and regulatory mechanisms suggest that they have evolved specialised roles tailored to the specific needs of *Plasmodium* proliferation.

We believe that ‘divergent’ is an appropriate descriptor for these kinases because it highlights their significant deviation from the canonical structures and functions of their counterparts in other organisms. These differences likely underlie the unique biology of *Plasmodium* and its adaptation to a complex life cycle in two host organisms. We have revised the manuscript to include an explanation of what we mean by ‘divergent’ in the context of these kinases (Lines 38-41).

3. It is conspicuous that all interacting partners of CDPK4 identified at the time point in which immunoprecipitation experiments were pursued were components of the replisome, given that CDPK4 is not primarily localized at the nucleus at that point. Instead, it is shown in Figure 1B to be localized to the cell poles. Are any of these interacting partners known to localize outside of the nucleus post-activation of gametocytes? Again, the authors discuss how CDPK4 might regulate its interacting partners by means of phosphorylation, however, direct phosphorylation of these putative substrates was not shown. They in fact mention that their data is in partial agreement with phosphoproteomic data previously obtained by other authors. Is it plausible that physical interactions with additional partners may play additional roles which haven't been discussed?

Among the identified interacting partners of CDPK4, kinesin-13 is known to be a protein located outside the nucleus in the *Plasmodium* male gametocyte (Zeeshan et al, 2022b). In addition, eIF3 is located in the cytoplasm in the asexual stage of *Plasmodium* (Dobrescu et al, 2023), but also expressed in gametocytes (Florens et al, 2002). In *Plasmodium*, ORC subunits are located in the nucleus (Mehra et al, 2005), but in other organisms, it is known that they also function while shuttling between cytoplasm and nucleus and are involved in cytokinesis or centriole duplication in the cytoplasm (Popova et al, 2018). Our live-cell imaging shows that CDPK4 is located in both nucleus and cytoplasm with strong focal points of concentration. Although we do not know whether the proteins immunoprecipitated with GFP-CDPK4 interact in the nucleus or the cytoplasm, it is possible that proteins with known major functions in the nucleus, such as ORC subunits, may interact with CDPK4 in either nucleus or cytoplasm.

CDPK4 is thought to function not only as a kinase but also as a scaffold protein, forming complexes with other proteins such as ORC subunits or CDC6 (Fang et al, 2017), and promoting the interaction between a phosphorylation product and its binding protein (Popova et al, 2018). As the reviewer points out, we show no direct phosphorylation in this study.

However, we believe that our interactome data for these three protein kinases suggest potential substrates of these kinases and candidate proteins that physically interact with and function in complex with these kinases. These suggestions can be followed up by future experimental analysis. We have revised the manuscript and added further relevant discussion (Lines 307-310 and 382-399).

4. Overall, the manuscript would benefit from a schematic summarizing all findings. It would be useful for the reader to get a general perspective of when and where these kinases are involved in the male gamete formation process, who they interact with and what evidence is gathered from other studies exploring phospho-changes in potential substrates as well as interacting partners. The model can also summarize how the authors envision these kinases "talk to each other" through the process.

We have included a schematic that summarises all the findings in Figure 5.

Minor points

5. Line 111 refers to the analysis of CDPK4 localization at >6 min time points, however, this is not shown (or where it is shown is not immediate neither is it specified).

We include images of male gametocytes at the >6min post activation time points, which have more than three foci of CDPK4-GFP as Figure S1D.

6. Line 114 specified that MAP2-GFP displays a nuclear localization. While this is clearly the case, it is also detectable in the cytosol, albeit not as markedly.

We agree with the reviewer's suggestion that MAP2-GFP is also detectable in the cytoplasm. We have revised this part of the manuscript (Lines 219-220).

Reviewer #3:

Summary

In this manuscript 3 previously studied kinases are revisited in a comparative fashion to flesh out where they differ in the endomitotic steps during male gametocytogenesis. This is important as timing of analysis is super-critical in this sec-min timescale process, which makes it hard to comparatively interpret studies from different laboratories, all using their own, different protocols. Besides the comparative angle, new insights are the pull-down proteomes for all 3 kinases, which partially overlap with previous phosphoproteomes - but this is a different level of data, which could be emphasizes more in the manuscript. Furthermore, expansion microscopy (U-ExM) is applied next to classic TEM. Overall, non-overlapping functions for the kinases are described as well as tantalizing hints as how chromosome condensation might be working in this process. Although the title suggests a focus on the MTOC, kinetochore and axoneme, the scope is beyond that by uncovering roles in ORCs, MCMs, and SMCs.

We appreciate the reviewer's recognition of the importance and novelty of our research. We believe that by widening and deepening the discussion of our findings, our arguments are clearer in the revised manuscript.

Specific points

My only pause is describing the pulled-down proteins with each kinase as substrates. They might be substrates but might be proteins in complex with substrates, or structural proteins used for spatial cues of the kinases. This should be interpreted more carefully.

We appreciate the reviewer's important comment. We recognise that the proteins identified by immunoprecipitation are putative interacting partners and should be referred to as such in the manuscript. They may be potential substrates but we provide no evidence to assess this possibility. We have revised the description of the identified proteins throughout the manuscript.

References

- Abe S, Kurata M, Suzuki S, Yamamoto K, Aisaki K ichi, Kanno J, Kitagawa M (2012) Minichromosome maintenance 2 bound with retroviral Gp70 is localized to Cytoplasm and enhances DNA-damage-induced apoptosis. *PLoS One* 7. doi:10.1371/journal.pone.0040129.
- Billker O, Lourido S, Sibley LD (2009) Calcium-Dependent Signaling and Kinases in Apicomplexan Parasites. *Cell Host Microbe* 5: 612–622. doi:10.1016/j.chom.2009.05.017.
- Dixit A, Singh PK, Sharma GP, Malhotra P, Sharma P (2010) PfSRPK1, a novel splicing-related kinase from Plasmodium falciparum. *Journal of Biological Chemistry* 285: 38315–38323. doi:10.1074/jbc.M110.119255.
- Dobrescu I, Hammam E, Dziekan JM, Claës A, Halby L, Preiser P, Bozdech Z, Arimondo PB, Scherf A, Nardella F (2023) Plasmodium falciparum Eukaryotic Translation Initiation Factor 3 is Stabilized by Quinazoline-Quinoline Bisubstrate Inhibitors. *ACS Infect Dis* 9: 1257–1266. doi:10.1021/acsinfectdis.3c00127.
- Fang H, Klages N, Baechler B, Hillner E, Yu L, Pardo M, Choudhary J, Brochet M (2017) Multiple short windows of calcium-dependent protein kinase 4 activity coordinate distinct cell cycle events during Plasmodium gametogenesis. *Elife* 6: 1–23. doi:10.7554/eLife.26524.001.
- Florens L, Washburn MP, Dale Raine J, Anthony RM, Grainger M, David Haynes J, Kathleen Moch J, Muster N, Sacci JB, Tabb DL *et al* (2002) A proteomic view of the Plasmodium falciparum life cycle.
- Invergo BM, Brochet M, Yu L, Choudhary J, Beltrao P, Billker O (2017) Sub-minute Phosphoregulation of Cell Cycle Systems during Plasmodium Gamete Formation. *Cell Rep* 21: 2017–2029. doi:10.1016/j.celrep.2017.10.071.
- Janse CJ, Franke-Fayard B, Waters AP (2006) Selection by flow-sorting of genetically transformed, GFP-expressing blood stages of the rodent malaria parasite, Plasmodium berghei. *Nat Protoc* 1: 614–623. doi:10.1038/nprot.2006.88.
- Kumar S, Gargaro OR, Kappe SHI (2022) Plasmodium falciparum CRK5 Is Critical for Male Gametogenesis and Infection of the Mosquito. *mBio* 13. doi:10.1128/mbio.02227-22.

- Mehra P, Biswas AK, Gupta A, Gourinath S, Chitnis CE, Dhar SK (2005) Expression and characterization of human malaria parasite *Plasmodium falciparum* origin recognition complex subunit 1. *Biochem Biophys Res Commun* 337: 955–966. doi:10.1016/j.bbrc.2005.09.131.
- Popova V V., Brechalov A V., Georgieva SG, Kopytova D V. (2018) Nonreplicative functions of the origin recognition complex. *Nucleus* 9: 460–473. doi:10.1080/19491034.2018.1516484.
- Rashpa R, Brochet M (2022) Expansion microscopy of *Plasmodium* gametocytes reveals the molecular architecture of a bipartite microtubule organisation centre coordinating mitosis with axoneme assembly. *PLoS Pathog* 18. doi:10.1371/journal.ppat.1010223.
- Tewari R, Dorin D, Moon R, Doerig C, Billker O (2005) An atypical mitogen-activated protein kinase controls cytokinesis and flagellar motility during male gamete formation in a malaria parasite. *Mol Microbiol* 58: 1253–1263. doi:10.1111/j.1365-2958.2005.04793.x.
- Zeeshan M, Brady D, Markus R, Vaughan S, Ferguson D, Holder AA, Tewari R (2022a) *Plasmodium* SAS4: basal body component of male cell which is dispensable for parasite transmission. *Life Sci Alliance* 5. doi:10.26508/lsa.202101329.
- Zeeshan M, Rashpa R, Ferguson DJP, Abel S, Chahine Z, Brady D, Vaughan S, Moores CA, Le Roch KG, Brochet M *et al* (2022b) Genome-wide functional analysis reveals key roles for kinesins in the mammalian and mosquito stages of the malaria parasite life cycle. *PLoS Biol* 20. doi:10.1371/journal.pbio.3001704.

March 6, 2025

RE: Life Science Alliance Manuscript #LSA-2024-03056R

Prof. Rita Tewari
University of Nottingham
School of Life Sciences
Queens Medical Centre
Nottingham NG7 2UH
United Kingdom

Dear Dr. Tewari,

Thank you for submitting your revised manuscript entitled "Divergent Plasmodium kinases drive MTOC, kinetochore and axoneme organisation in male gametogenesis". We would be happy to publish your paper in Life Science Alliance pending final revisions necessary to meet our formatting guidelines.

- please be sure that the authorship listing and order is correct
- please add ORCID ID for the secondary corresponding author- they should have received instructions on how to do so
- please be sure that all authors are mentioned in the Authors' Contributions section
- the contributions selected for Anthony Holder do not qualify them for authorship. Please either update the contributions in our system and the Author Contributions section of the manuscript or let us know if the author needs to be removed (and added eventually to the acknowledgment section)
- please move your main, supplementary figure, and table legends to the main manuscript text after the references section
- we encourage you to revise the figure legends for figure S2 such that the figure panels are introduced in alphabetical order
- please add callouts for Figures 2B; 5 and S4A, B to your main manuscript text

LSA now encourages authors to provide a 30-60 second video where the study is briefly explained. We will use these videos on social media to promote the published paper and the presenting author (for examples, see <https://docs.google.com/document/d/1-UWCfbE4pGcDdcgzcmiuJl2XMBJnxKYeqRvLLrLSo8s/edit?usp=sharing>). Corresponding or first-authors are welcome to submit the video. Please submit only one video per manuscript. The video can be emailed to contact@life-science-alliance.org

A. FINAL FILES:

B. MANUSCRIPT ORGANIZATION AND FORMATTING:

Sincerely,

March 11, 2025

RE: Life Science Alliance Manuscript #LSA-2024-03056RR

Prof. Rita Tewari
University of Nottingham
School of Life Sciences
Queens Medical Centre
Nottingham NG7 2UH
United Kingdom

Dear Dr. Tewari,

Thank you for submitting your Research Article entitled "Divergent Plasmodium kinases drive MTOC, kinetochore and axoneme organisation in male gametogenesis". It is a pleasure to let you know that your manuscript is now accepted for publication in Life Science Alliance. Congratulations on this interesting work.

DISTRIBUTION OF MATERIALS:

Again, congratulations on a very nice paper. I hope you found the review process to be constructive and are pleased with how the manuscript was handled editorially. We look forward to future exciting submissions from your lab.

Sincerely,
